# X-chromosome target specificity diverged between dosage compensation mechanisms of two closely related Caenorhabditis species

**Qiming Yang[1,2†], Te-Wen Lo[1,2†‡], Katjuša Brejc[1,2], Caitlin Schartner[1,2§], Edward J Ralston[1,2], Denise M Lapidus[1,2], Barbara J Meyer[1,2]***

[1]Howard Hughes Medical Institute, Berkeley, United States; [2]Department of Molecular and Cell Biology, University of California, Berkeley, Berkeley, United States

**\*For correspondence:**
bjmeyer@berkeley.edu

[†]These authors contributed equally to this work

**Present address:** [‡]Department of Biology, Ithaca College, Ithaca, United States; [§]Roche Diagnostics, Santa Clara, Canada

**Abstract** An evolutionary perspective enhances our understanding of biological mechanisms. Comparison of sex determination and X-chromosome dosage compensation mechanisms between the closely related nematode species *Caenorhabditis briggsae* (*Cbr*) and *Caenorhabditis elegans* (*Cel*) revealed that the genetic regulatory hierarchy controlling both processes is conserved, but the X-chromosome target specificity and mode of binding for the specialized condensin dosage compensation complex (DCC) controlling X expression have diverged. We identified two motifs within *Cbr* DCC recruitment sites that are highly enriched on X: 13 bp MEX and 30 bp MEX II. Mutating either MEX or MEX II in an endogenous recruitment site with multiple copies of one or both motifs reduced binding, but only removing all motifs eliminated binding in vivo. Hence, DCC binding to *Cbr* recruitment sites appears additive. In contrast, DCC binding to *Cel* recruitment sites is synergistic: mutating even one motif in vivo eliminated binding. Although all X-chromosome motifs share the sequence CAGGG, they have otherwise diverged so that a motif from one species cannot function in the other. Functional divergence was demonstrated in vivo and in vitro. A single nucleotide position in *Cbr* MEX can determine whether *Cel* DCC binds. This rapid divergence of DCC target specificity could have been an important factor in establishing reproductive isolation between nematode species and contrasts dramatically with the conservation of target specificity for X-chromosome dosage compensation across *Drosophila* species and for transcription factors controlling developmental processes such as body-plan specification from fruit flies to mice.

## Editor's evaluation

This important study uses state-of-the-art methods to explore the evolution of dosage compensation between two closely related nematode species. The evidence supporting the rapid evolution of the recruitment motifs on the X chromosome, despite a general conservation of the dosage compensation machinery, is compelling. This work will be of broad interest to cell biologists and evolutionary biologists.

## Introduction

Comparative studies have shown that different facets of metazoan development exhibit remarkably different degrees of conservation across species (*Carroll, 2008*). At one extreme, homeobox-containing *Hox* genes and *Wnt*-pathway signaling genes play conserved roles in body plan formation (*Hox*) and cell-fate determination, neural patterning, or organogenesis (*Wnt*) across clades diverged

by more than 600 million years (MYR) (*Malicki et al., 1990*; *De Kumar and Darland, 2021*; *Rim et al., 2022*). Distant orthologous genes within these ancestral pathways can substitute for each other. For example, both the mouse *Small eye* (*Pax-6*) gene (*Hill et al., 1991*) and the fruit fly *eyeless* (*ey*) gene (*Quiring et al., 1994*; *Halder et al., 1995*) control eye morphogenesis and encode a transcription factor that includes a paired domain and a homeodomain. Ectopic expression of mouse *Pax-6* in different fruit fly imaginal disc primordia can induce morphologically normal ectopic compound eye structures on fruit fly wings, legs, and antennae (*Halder et al., 1995*). Hence, at a deep level, eye morphogenesis is under related genetic and molecular control in vertebrates and insects, despite profound differences in eye morphology and mode of development.

At the other extreme are aspects of development related to sex. For example, chromosomal strategies to determine sexual fate in mice, fruit flies, and nematodes (XY or XO males and XX females or hermaphrodites) and the mechanism needed to compensate for the consequent difference in X-chromosome dose between sexes have diverged greatly. To balance X gene expression between sexes, female mice randomly inactivate one X chromosome (*Yin et al., 2021*; *Loda et al., 2022*), while male fruit flies double expression of their single X chromosome (*Samata and Akhtar, 2018*; *Rieder et al., 2019*), and hermaphrodite worms halve expression of both X chromosomes (*Meyer, 2022a*; *Meyer, 2022b*).

The divergence in these pathways is so great that comparisons among animals of the same genus can provide useful evolutionary context for understanding the developmental mechanisms that distinguish the sexes. Therefore, we determined the genetic and molecular specification of sexual fate and X-chromosome dosage compensation in the nematode *C. briggsae* and compared it to the wealth of knowledge amassed about these processes in *C. elegans*. These two species have diverged by 15–30 MYR (*Cutter, 2008*).

In *C. elegans*, the sex determination and dosage compensation pathways are linked by genes that coordinately control both processes. For example, in XX embryos, the switch gene *sdc-2* sets the sex determination pathway to the hermaphrodite mode and triggers the binding of a DCC onto both X chromosomes to reduce X gene expression by half and thereby match X expression with that from XO males (*Meyer, 2022a*). The DCC shares subunits with condensin, a protein complex that controls the structure, resolution, and segregation of mitotic and meiotic chromosomes from yeast to humans (*Yatskevich et al., 2019*; *Meyer, 2022b*).

We determined the extent to which the sex-specific gene regulatory hierarchy is conserved between *C. elegans* and *C. briggsae* and the extent to which subunits of the *C. briggsae* DCC correspond to those of the *C. elegans* DCC. We also defined the *cis*-acting regulatory sites that confer X-chromosome specificity and recruit the *C. briggsae* DCC. We found that the DCC itself and the regulatory hierarchy that determines sex and directs the DCC to X have been conserved, but remarkably, both the X-chromosome target specificity of the *C. briggsae* DCC and its mode of binding to X have diverged.

## Results

### Conservation between *C. briggsae* and *C. elegans* of the core dosage compensation machinery and genetic hierarchy that regulates dosage compensation

The pivotal hermaphrodite-specific regulatory protein that coordinately controls both sex determination and dosage compensation in *C. elegans* is a 350 kDa protein called SDC-2. It directs the DCC to both X chromosomes of XX embryos to achieve dosage compensation and also activates the hermaphrodite program of sexual differentiation (*Chuang et al., 1996*; *Dawes et al., 1999*; *Chu et al., 2002*; *Pferdehirt et al., 2011*). Loss of *Cel sdc-2* causes XX-specific lethality due to excessive X-chromosome gene expression and masculinization of escaper animals (*Nusbaum and Meyer, 1989*; *Kruesi et al., 2013*).

SDC-2 has no known homologs outside of nematodes and only a coiled-coil domain as a predicted structural feature (*Meyer, 2022a*). Among five *Caenorhabditis* species compared, the entire SDC-2 protein has 23–29% identity and 38–45% similarity (*Figure 1—figure supplement 2A*). Between *Cbr* and *Cel*, the entire SDC-2 protein shows 26% identity and 43% similarity (*Figure 1—figure*

*supplements 1 and 2A*). To assess the conservation of gene function, we deployed genome-editing technology in *C. briggsae* to knockout *sdc-2*.

Using a PCR-based molecular strategy to identify insertions and deletions induced by DNA repair following directed mutagenesis with zinc finger nucleases, we recovered several independent *Cbr sdc-2* mutant lines (*Figure 1—figure supplement 3*). Homozygous *Cbr sdc-2* mutations caused extensive XX-specific lethality, consistent with a defect in dosage compensation and the conservation of gene function (*Figure 1A*). Nearly all *Cbr sdc-2* hermaphrodites died as embryos or young larvae; rare XX survivors exhibited slow growth and masculinization. *Cbr sdc-2* males were viable (*Figure 1A*) and had wild-type body morphology.

To determine whether the hermaphrodite-specific lethality of *Cbr sdc-2* mutants was caused by defects in dosage compensation, we first identified components of the *C. briggsae* DCC and then asked whether DCC binding to X is disrupted by mutation of *Cbr sdc-2*, as it is by mutation of *Cel sdc-2*. In *C. elegans*, five of the ten known DCC proteins are homologous to subunits of condensin, an evolutionarily conserved protein complex required to restructure and resolve chromosomes in preparation for cell divisions in mitosis and meiosis (*Figure 1B*; *Chuang et al., 1994*; *Lieb et al., 1996*; *Lieb et al., 1998*; *Chan et al., 2004*; *Tsai et al., 2008*; *Csankovszki et al., 2009*; *Mets and Meyer, 2009*; *Yatskevich et al., 2019*; *Meyer, 2022a*). The evolutionary time scale over which condensin subunits were co-opted for dosage compensation in nematodes had not been explored.

Several lines of evidence indicate that a condensin complex mediates dosage compensation in *C. briggsae* as well. First, BLASTP searches revealed *C. briggsae* orthologs of all known *C. elegans* DCC condensin subunits (*Figure 1B*). Alignment of DPY-27 protein revealed 38% identity and 56% similarity between *C. elegans* and *C. briggsae* (*Figure 1—figure supplement 2B*). Immunofluorescence experiments using antibodies against *Cbr* DPY-27, the SMC4 ortholog of the only *Cel* DCC condensin subunit (*Cel* DPY-27) not associated with mitotic or meiotic condensins (*Chuang et al., 1994*), revealed X chromosome-specific localization in hermaphrodites, but not males, indicating conservation of function (*Figure 1C* and *Figure 2A and B*). Specificity of DPY-27 antibodies was demonstrated by Western blot analysis (*Figure 1—figure supplement 4A*).

Second, disruption of *Cbr dpy-27* conferred hermaphrodite-specific lethality, with rare XX escaper animals exhibiting a dumpy (Dpy) phenotype, like the disruption of *Cel dpy-27* (*Figure 1G*). Immunofluorescence experiments with *Cbr* DPY-27 antibodies revealed diffuse nuclear distribution of DPY-27 in Dpy escapers of *dpy-27(y436)* mutants instead of X localization, consistent with lethality (*Figure 1D*).

Third, co-immunoprecipitation of proteins with rabbit *Cbr* DPY-27 antibodies followed by SDS-PAGE and mass spectrometry of excised trypsinized protein bands identified *Cbr* MIX-1 (*Table 1*; Materials and methods), the SMC2 condensin subunit ortholog found in the *Cel* DCC complex (*Lieb et al., 1998*; *Figure 1B*). Both DPY-27 and MIX-1 belong to the SMC family of chromosomal ATPases that dimerize and participate in condensin complexes (*Figure 1B*).

Fourth, immunofluorescence experiments using *Cbr* MIX-1 antibodies (*Figure 1—figure supplement 4B*) revealed co-localization of *Cbr* MIX-1 with *Cbr* DPY-27 on hermaphrodite X chromosomes (*Figure 1E*). *Cbr* MIX-1 protein did not bind to X chromosomes in *Cbr dpy-27(y436)* mutant animals (*Figure 1F*). Instead, MIX-1 exhibited diffuse nuclear distribution, like DPY-27, consistent with the two proteins participating in a complex and the dependence of MIX-1 on DPY-27 for its binding to X (*Figure 1F*). These data demonstrate that condensin subunits play conserved roles in the dosage compensation machinery of both *C. briggsae* and *C. elegans*.

In contrast to DPY-27, MIX-1 shows 55% identity and 72% similarity between *C. elegans* and *C. briggsae*. Not only does MIX-1 participate in the DCC, it also participates in two other distinct *Caenorhabditis* condensin complexes that are essential for the proper resolution and segregation of mitotic and meiotic chromosomes (*Mets and Meyer, 2009*; *Csankovszki et al., 2009*). Conserved roles in chromosome segregation complexes would constrain MIX-1 sequence divergence, thereby explaining its greater conservation between species.

Evidence that DCC binding defects underlie the XX-specific lethality caused by *Cbr sdc-2* mutations is our finding that neither *Cbr* DPY-27 (*Figure 2C*) nor *Cbr* MIX-1 (not shown) binds to X chromosomes in *Cbr sdc-2* mutant hermaphrodites. Instead, we found a low level of diffuse nuclear staining. Thus, the role of *sdc-2* in the genetic hierarchies that activate dosage compensation is also conserved.

We next explored why maternally supplied DCC subunits fail to bind to the single X chromosome of *C. briggsae* males. In *C. elegans* XO embryos, the master switch gene *xol-1* (XO lethal) represses

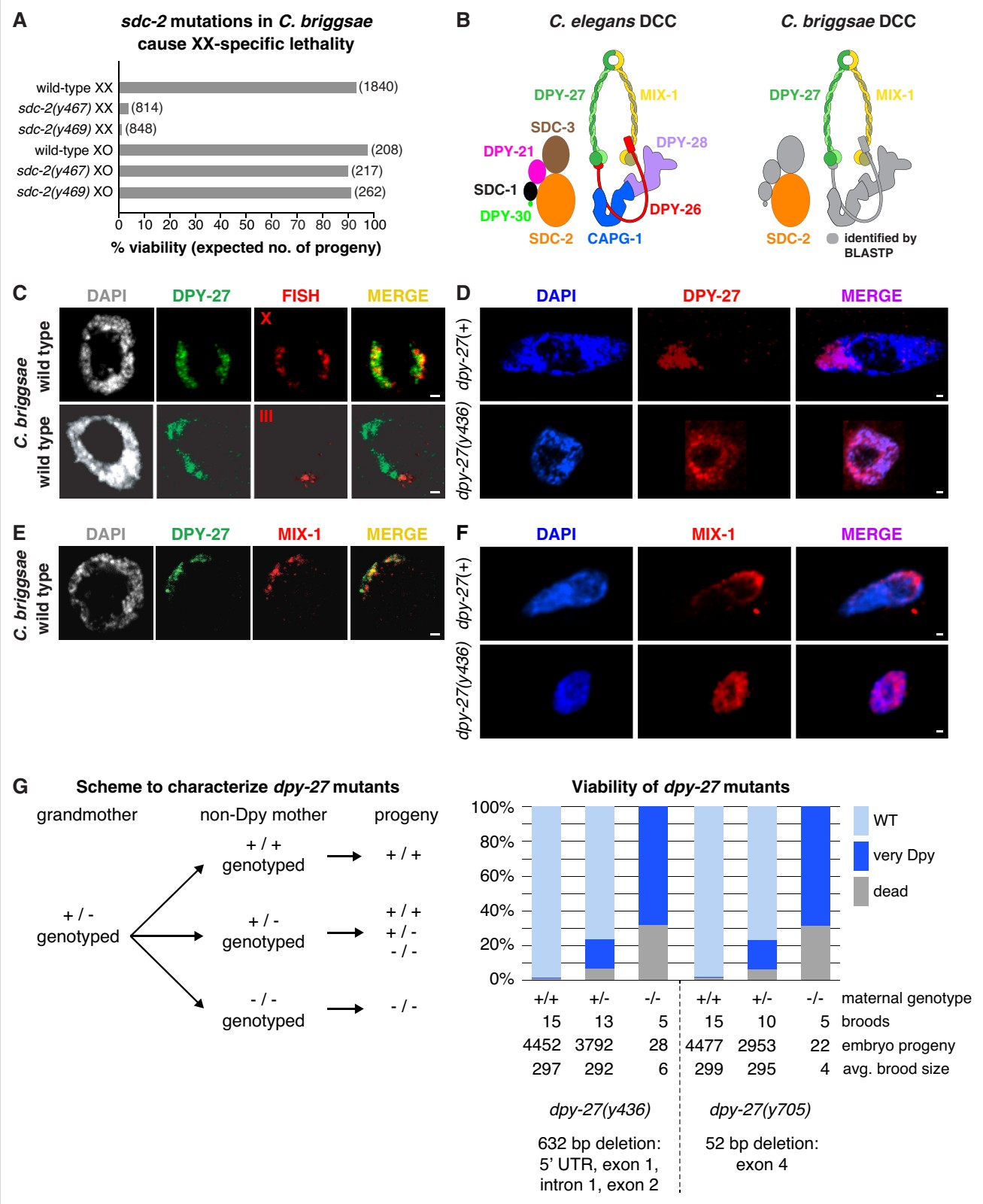

**Figure 1.** Conservation of X-chromosome dosage compensation machinery between *C. briggsae* and *C. elegans*. (**A**) *sdc-2* mutations cause XX-specific lethality in *C. briggsae*. Graph shows percent viability of wild-type and *Cbr sdc-2* mutant XX and XO adults. Viability of homozygous XX and hemizygous XO *Cbr sdc-2* mutants is expressed as the percentage of live adults for each karyotype relative to the number expected (shown in parentheses) in the progeny of a cross if all mutant animals were viable. Crosses and calculations are described in Materials and methods. Sequence changes of

*Figure 1 continued on next page*

Figure 1 continued

*sdc-2* mutations derived from genome editing using zinc-finger nucleases are shown in ***Figure 1—figure supplement 3A***. (**B**) Schematic of the *C. elegans* dosage compensation complex (left) and *C. briggsae* orthologs identified by BLASTP (right). The *C. elegans* dosage compensation complex (DCC) includes homologs of all core condensin subunits (MIX-1, DPY-27, DPY-26, DPY-28, and CAPG-1). *C. briggsae* DCC components identified and characterized in this study are shown in color; other orthologs are in gray. DPY-27 and MIX-1 belong to the SMC (S̲tructural M̲aintenance of C̲hromosomes) family of chromosomal ATPases. Each has nucleotide-binding domains (NBDs) at its N- and C-termini that are linked by two long coiled-coil domains separated by a hinge domain. Each SMC protein folds back on itself to form a central region of two anti-parallel coiled coils flanked by the NBDs and the hinge. DPY-27 and MIX-1 dimerize through interactions between their hinge domains and their NBD domains. The globular NBDs bind to the three non-SMC condensin DCC subunits (DPY-26, DPY-28, and CAPG-1) (See ***Meyer, 2022a***). (**C**) Condensin subunit DPY-27 binds X chromosomes and mediates dosage compensation in *C. briggsae*. Confocal images of *C. briggsae* hermaphrodite gut nuclei co-stained with the DNA dye DAPI (gray), antibodies to *Cbr* DPY-27 (green), and FISH probes to either 5% of X (red, top), or 1% of chromosome III (red, bottom) show that *Cbr* DPY-27 co-localizes with X but not III, consistent with a role in dosage compensation. Scale bars, 1 μm. (**D**) Confocal images of *C. briggsae* gut nuclei from *dpy-27*(+) or *dpy-27*(*y436*) mutant XX adult hermaphrodites co-stained with DAPI (blue) and the *Cbr* DPY-27 rabbit antibody (red). DPY-27 shows subnuclear localization in a *dpy-27*(+) gut nucleus (top), as expected for X localization. The mutant gut nucleus (bottom) shows diffuse nuclear distribution of DPY-27, as anticipated for a mutant SMC-4 condensin ortholog that lacks most of the N-terminal part of the ATPase domain and, therefore, has no ATP binding or hydrolysis. Scale bars, 1 μm. (**E**) Confocal images of a *C. briggsae* gut nucleus from wild-type adult hermaphrodites co-stained with DAPI (gray) and antibodies to *Cbr* DPY-27 (green) and *Cbr* MIX-1 (red) show that *Cbr* MIX-1 co-localizes with *Cbr* DPY-27 on X in wild-type hermaphrodites. Scale bars, 1 μm. (**F**) Association of *Cbr* MIX-1 (red) with X found in a *dpy-27*(+) nucleus (top) is disrupted in a *Cbr dpy-27*(*y436*) nucleus (bottom), in accord with participation of *Cbr* MIX-1 in a protein complex with *Cbr* DPY-27. Scale bars, 1 μm. (**G**) Viability of *dpy-27* mutant XX *C. briggsae* animals. The left panel shows the genetic scheme to characterize the effect of maternal genotype on viability of *dpy-27* null XX mutants. Comparison is made between homozygous null *dpy-27* progeny from heterozygous or homozygous non-Dpy mutant mothers. The genotype of non-DPY mothers was established through PCR analysis. The right panel shows the percent viability of progeny from wild-type hermaphrodites and heterozygous or homozygous *dpy-27* mutant hermaphrodites. The maternal genotype, number of broods, total number of embryo progeny from all broods, and average brood size are provided for two null alleles of *dpy-27*. Molecular characterization of mutations is shown below the graph and in ***Figure 1—figure supplement 3B***. Almost all progeny of *dpy-27* null mutant mothers are dead; a homozygous *dpy-27* null strain cannot be propagated. More than 20% of progeny of *dpy-27*/+heterozygous mutant mothers are very Dpy or dead, indicating that a wild-type DPY-27 maternal contribution has minimal effect on suppressing the deleterious effect of the homozygous null zygotic genotype. The complete XX lethality is consistent with a major role for condensin subunit DPY-27 in dosage compensation.

The online version of this article includes the following source data and figure supplement(s) for figure 1:

**Figure supplement 1.** Protein sequence alignment comparing SDC-2 proteins in *C. elegans* and *C. briggsae*.

**Figure supplement 2.** Conservation of SDC-2 and DPY-27 proteins in the *Caenorhabditis* genus.

**Figure supplement 3.** DNA sequence changes mediated by genome editing.

**Figure supplement 4.** Specificity of *Cbr* DPY-27 and MIX-1 antibodies.

**Figure supplement 4—source data 1.** Source data for DPY-27 and MIX-1 antibody specificity.

the hermaphrodite-specific *sdc-2* gene required for DCC binding to X and thereby prevents other DCC subunits from functioning in males (***Miller et al., 1988***; ***Rhind et al., 1995***; ***Dawes et al., 1999***; ***Meyer, 2022a***). Loss of *Cel xol-1* activates *Cel sdc-2* in XO embryos, causing DCC binding to X, reduction in X-chromosome gene expression, and consequent death. We isolated the null mutant allele *Cbr xol-1*(*y430*) by PCR screening of a *C. briggsae* deletion library (***Supplementary file 1***). We found that the *Cbr xol-1* mutation caused inappropriate binding of the DCC to the single X of XO embryos (***Figure 2D***) and fully penetrant male lethality (***Figure 3B***), as expected from the disruption of a gene that prevents the DCC machinery from functioning in *C. briggsae* males. *Cbr xol-1* mutant XX hermaphrodites appeared wild-type.

To investigate the hierarchical relationship between *Cbr xol-1* and *Cbr sdc-2*, we asked whether a *Cbr sdc-2* mutation could suppress the male lethality caused by a *Cbr xol-1* mutation. Both genes are closely linked in *C. briggsae*, prompting us to use genome editing technology to introduce de novo mutations in cis to pre-existing lesions without relying on genetic recombination between closely linked genes. If *Cbr xol-1* controls *Cbr sdc-2*, then mutation of *Cbr sdc-2* should rescue the male lethality of *Cbr xol-1* mutants (***Figure 2E***). This prediction proved to be correct. XO males were observed among F1 progeny from mated *Cbr xol-1* hermaphrodites injected with ZFNs targeting *Cbr sdc-2* (***Figure 3A, B and D***). Insertion and deletion mutations were found at the *Cbr sdc-2* target site in more than twenty tested F1 males (examples are in ***Figure 1—figure supplement 3C and D***). Quantification of male viability in four different *xol-1 sdc-2* mutant lines revealed nearly full rescue (***Figure 3B***), with a concomitant absence of DCC binding on the single X chromosome (***Figure 2E***). Therefore, *Cbr xol-1* functions upstream of *Cbr sdc-2* to repress it and thereby prevents DCC binding to the male X

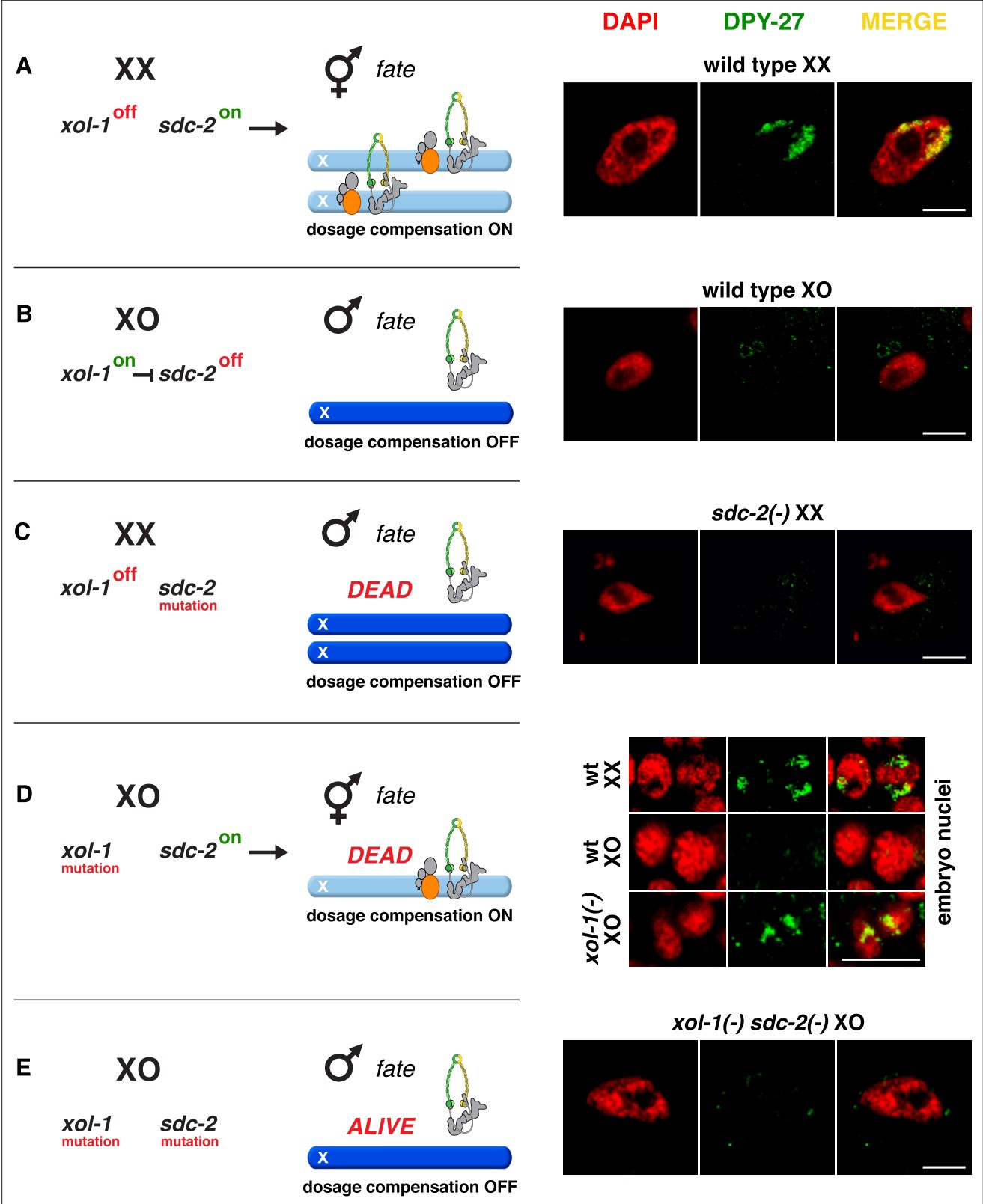

**Figure 2.** Conserved genetic hierarchy targets the *C. briggsae* dosage compensation complex (DCC) to the X chromosomes of hermaphrodites. (**A–E**) Schematic depiction of the genetic hierarchy controlling sex-specific DCC recruitment to *C. briggsae* X chromosomes (left) paired with representative immunofluorescence experiments exemplifying DCC localization (right). Scale bars, 5 µm. Gut nuclei (**A, B, C, E**) or embryos (**D**) were co-stained with DAPI (red) and antibodies to *Cbr* DPY-27 (green). In wild-type XX, but not XO gut nuclei (**A, B**), DPY-27 co-localizes with X chromosomes, consistent

*Figure 2 continued on next page*

*Figure 2 continued*

with a role for condensin subunit DPY-27 in dosage compensation (see also *Figure 1C*). (**C**) SDC-2 is required for recruitment of DPY-27 to the X chromosomes of hermaphrodites. Failure of the DCC to bind X chromosomes of *sdc-2* XX mutants underlies the XX-specific lethality. Shown is the gut nucleus of a rare XX *sdc-2* mutant escaper near death. *sdc-2* mutant XX escaper animals are masculinized. (**D**) Lethality of *Cbr xol-1(y430)* XO animals corresponds to inappropriate binding of the DCC to the single X in embryos. (**E**) Mutation of the DCC recruitment factor *Cbr sdc-2* in a *Cbr xol-1* XO mutant prevents DCC recruitment to X and suppresses the XO lethality. See *Figure 3B* for quantification.

chromosome. In summary, not only is the core condensin dosage compensation machinery conserved between *Caenorhabditis* species, but so also are the key features of the genetic hierarchy that confers sex-specificity to the dosage compensation process.

## Conservation between *C. briggsae* and *C. elegans* of the genetic hierarchy that regulates early stages of sex determination

Mechanisms controlling sex determination and differentiation are dynamic over evolutionary time; major differences can exist even within an individual species. For example, males within the house fly species *Musca domestica* can utilize one of many different male-determining factors on autosomes and sex chromosomes to determine sex depending on a factor's linkage to other beneficial traits (*Meisel et al., 2016*).

Within the *Caenorhabditis* genus, similarities and differences occur in the genetic pathways governing the later stages of sex determination and differentiation (*Haag, 2005*). For example, three sex-determination genes required for *C. elegans* hermaphrodite sexual differentiation but not dosage compensation, the transformer genes *tra-1*, *tra-2*, and *tra-3*, are conserved between *C. elegans* and *C. briggsae* and play very similar roles. Mutation of any one gene causes virtually identical masculinizing somatic and germline phenotypes in both species (*Kelleher et al., 2008*). Moreover, the DNA binding motif for both *Cel* and *Cbr* TRA-1 (*Berkseth et al., 2013*), a Ci/GL1 zinc-finger transcription factor that acts as the terminal regulator of somatic sexual differentiation (*Zarkower and Hodgkin, 1992*), is conserved between the two species.

At the opposite extreme, the mode of sexual reproduction, hermaphroditic versus male/female, dictated the genome size and reproductive fertility of *Caenorhabditis* species diverged by only

**Table 1.** MALDI-TOF identification of *Cbr* MIX-1 peptides.

| m/z Submitted | MH+ Matched | Delta ppm | Peptide | Missed Cleavage | Database Sequence |
|---|---|---|---|---|---|
| 916.47 | 916.46 | 9.5 | 674–680 | 0 | (K)YHENVVR(L) |
| 1163.59 | 1163.58 | 3.3 | 375–384 | 1 | (K)LRGELEGMSR(G) |
| 1214.65 | 1214.66 | −3.6 | 631–641 | 0 | (R)VLIESQCLPGR(R) |
| 1224.63 | 1224.62 | 8.8 | 713–723 | 1 | (R)EVAYTDGVKSR(T) |
| 1263.74 | 1263.74 | −0.87 | 524–534 | 0 | (R)DVEGLVLHLIR(L) |
| 1285.69 | 1285.69 | −2.8 | 631–641 | 0 | (R)VLIESQCLPGR(R) |
| 1350.69 | 1350.70 | −8.9 | 656–666 | 0 | (R)YTIINDQSLQR(A) |
| 1881.97 | 1881.98 | −2.3 | 134–150 | 0 | (R)GVGLNVNNPHFLIMQGR(I) |
| 1886.89 | 1886.91 | −6.8 | 86–101 | 0 | (K)QSPFGMDHLDELVVQR(H) |
| 2064.01 | 2064.00 | 3.4 | 460–477 | 0 | (K)ITQQVQSLGYNADEDVQR(R) |
| 2377.18 | 2377.16 | 5.6 | 385–415 | 1 | (R)GTVTNDKGEHVSLETYIQETR(A) |

This table lists the mass-to-charge ratio (m/z) of measured peptides, the predicted masses (MH+ Matched), and the deviation from predicted masses (Delta ppm). The ID of each measured peptide is described by the residue range within full-length MIX-1 (Peptide) and its corresponding amino acid sequence (Database Sequence). The number of uncut tryptic peptide bonds is listed for each peptide (Missed Cleavage).

In addition to MIX-1, MALDI-TOF analysis of excised protein bands in the molecular weight range of condensin subunits excised from an SDS-PAGE gel revealed peptides corresponding to four common high-molecular weight contaminants: the three vitellogenin yolk proteins VIT-2, VIT-4, VIT-5, and CBG14234, an ortholog of VIT-4. No protein bands corresponding to the molecular weights of SDC-2 or SDC-3 were visible on the SDS-PAGE gel.

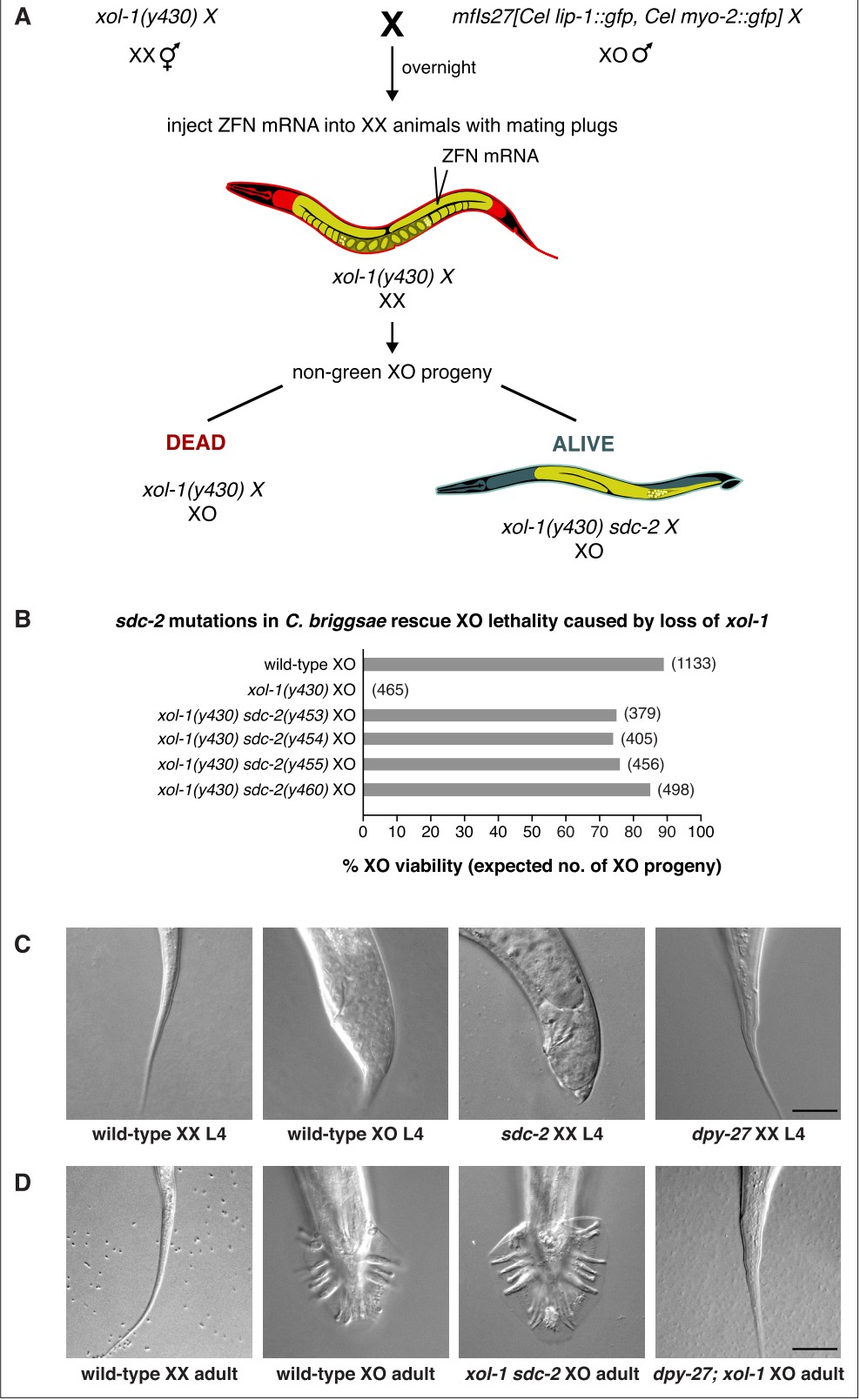

**Figure 3.** *sdc-2* controls dosage compensation and sex determination in *C. briggsae*. (**A**) Diagram of the screening strategy to recover *Cbr sdc-2* mutations as suppressors of the XO-specific lethality caused by a *xol-1* mutation. *Cbr xol-1* XX hermaphrodites were mated with males carrying a *gfp*-marked X chromosome to allow F1 XO males to be monitored for the parental origin of the X chromosome. Animals with mating plugs (indicating

*Figure 3 continued on next page*

*Figure 3 continued*

successful mating) were injected with mRNAs to *sdc-2* zinc-finger nucleases, and all F1 males were examined for GFP fluorescence. Non-green males necessarily inherited an X chromosome carrying a *Cbr-xol-1* mutation and, assuming conservation of the dosage compensation complex (DCC) regulatory hierarchy, would be inviable without a concomitant *Cbr sdc-2* mutation. GFP-positive males arose at low frequency from fertilization of nullo-X oocytes (caused by non-disjunction of the maternal X chromosome) with *gfp*-X-bearing sperm. These false positives were discarded from further study. (**B**) *Cbr sdc-2* mutations rescue *Cbr xol-1(y430)* XO lethality. Graph shows percent viability of wild-type XO animals and mutant XO animals carrying combinations of *Cbr xol-1* and *Cbr sdc-2* mutations. The % XO viability is expressed as the percentage of live XO adults relative to the number expected (shown in parentheses) in the progeny of the cross. Formulae for viability calculations are given in the Materials and methods. Sequence changes of *sdc-2* mutations are shown in ***Figure 1—figure supplement 3C and D***. (**C**) *sdc-2* activates the program for C*br* hermaphrodite sexual development. DIC images show the comparison of tail morphologies for *Cbr* L4 animals of different genotypes. *sdc-2* mutations, but not *dpy-27* mutations, cause masculinization of XX animals. Scale bar, 20 µm. (**D**) DIC images show tail morphologies of wild-type or doubly mutant *Cbr* adults. An *sdc-2* mutation suppresses both the XO lethality and feminization caused by a *xol-1* mutation, consistent with a role for *sdc-2* in controlling both dosage compensation and sex determination. *xol-1 sdc-2* XO animals are viable, fertile males, indicating that the *sdc-2* mutation suppressed the lethality and feminization caused by *xol-1* mutations in XO animals. A *dpy-27* mutation suppresses the XO lethality but not feminization caused by a *xol-1* mutation, consistent with a role for *dpy-27* in dosage compensation but not sex determination. *dpy-27; xol-1* XO animals are fertile hermaphrodites. Scale bar, 20 µm.

3.5 million years (***Yin et al., 2018***; ***Cutter et al., 2019***). Species that evolved self-fertilization (e.g. *C. briggsae* or *C. elegans*) lost 30% of their DNA content compared to male/female species (e.g. *C. nigoni* or *C. remanei*), with a disproportionate loss of male-biased genes, particularly the *male secreted short* (*mss*) gene family of sperm surface glycoproteins (***Yin et al., 2018***). The *mss* genes are necessary for sperm competitiveness in male/female species and are sufficient to enhance it in hermaphroditic species. Thus, sex has a pervasive influence on genome content.

In contrast to these later stages of sex determination and differentiation, the earlier stages of sex determination and differentiation had not been analyzed in *C. briggsae*. Therefore, we asked whether *xol-1* and *sdc-2* control sexual fate as well as dosage compensation in *C. briggsae*, as they do in *C. elegans*, over the 15–30 MYR that separates them. Our analysis of *Cbr sdc-2* XX mutant phenotypes revealed intersexual tail morphology in the rare animals that survived to the L3/L4 stage (***Figure 3C***), indicating a role for *Cbr sdc-2* in sex determination. Sexual transformation to the male fate was unlikely to have resulted from a disruption in dosage compensation since such transformation was never observed in *Cbr dpy-27* XX mutants (***Figure 3C***). Analysis of sexual phenotypes in double mutant strains confirmed that *Cbr sdc-2* controls sex determination. Specifically, *Cbr xol-1 Cbr sdc-2* double mutant XO animals develop as males, whereas *Cbr dpy-27; Cbr xol-1* double mutant XO animals develop as hermaphrodites (***Figure 3C and D***). That is, both *Cbr sdc-2* and *Cbr dpy-27* mutations suppress the XO lethality caused by a *xol-1* mutation, but only *Cbr sdc-2* mutations also suppress the sexual transformation of XO animals into hermaphrodites. These results show that both *sdc-2* and *dpy-27* function in *C. briggsae* dosage compensation, but only *sdc-2* also functions in sex determination. Thus, the two master regulatory genes that control the earliest stages of both sex determination and X-chromosome dosage compensation, *xol-1* and *sdc-2*, are conserved between *C. briggsae* and *C. elegans*.

## DCC recruitment sites isolated from *C. briggsae* X chromosomes fail to bind the *C. elegans* DCC

Discovery that the dosage compensation machinery and the gene regulatory hierarchy that controls sex determination and dosage compensation are functionally conserved between *C. briggsae* and *C. elegans* raised the question of whether the *cis*-acting regulatory sequences that recruit dosage compensation proteins to X chromosomes are also conserved. In *C. elegans*, the DCC binds to recruitment elements on X (*rex*) sites and then spreads across X to sequences lacking autonomous recruitment ability (***Csankovszki et al., 2004***; ***Jans et al., 2009***; ***Pferdehirt et al., 2011***; ***Albritton et al., 2017***; ***Anderson et al., 2019***). Within *rex* sites, combinatorial clustering of three DNA sequence motifs directs synergistic binding of the DCC (***Fuda et al., 2022***). To compare X-recruitment mechanisms between species, DNA binding sites for the *Cbr* DCC recruitment protein SDC-2 and the

*Cbr* DCC condensin subunit DPY-27 were defined by chromatin immuno-precipitation experiments followed by sequencing of captured DNA (ChIP-seq experiments) (*Figure 4A*). SDC-2 sites were obtained with anti-FLAG antibodies from a genome-engineered *Cbr* strain encoding a FLAG-tagged version of endogenous SDC-2. DPY-27 sites were obtained from either a wild-type *Cbr* strain with DPY-27 antibodies or from a genome-engineered strain encoding endogenous FLAG-tagged DPY-27 with anti-FLAG antibodies.

A consistent set of twelve large, overlapping SDC-2 ChIP-seq peaks and DPY-27 ChIP-seq peaks emerged from the studies (*Figure 4A*), representing less than one-fourth the number of DCC peaks than on the *C. elegans* X chromosome, which is smaller (17.7 Mb for *Cel* vs. 21.5 Mb for *Cbr*). SDC-2 and DPY-27 binding to autosomes was indistinguishable from that of the IgG control (*Figure 4— figure supplement 1A and B*). To determine whether DNA from these peaks acts as autonomous recruitment sites that confer X-chromosome target specificity to the dosage compensation process, we conducted DCC recruitment assays in vivo (*Figure 4B*). Assays were modeled on *rex* assays developed for *C. elegans* (Materials and methods and *Fuda et al., 2022*). Embryos carrying extrachromosomal arrays composed of multiple copies of DNA from a single ChIP-seq peak were stained with DPY-27 antibodies and a FISH probe to the array. DPY-27 localized to 80–90% of extrachromosomal arrays carrying DNA from each of the individual peaks (*Figure 4C and E* and *Table 2A*). In contrast, extrachromosomal arrays made from three regions of X lacking DCC binding in ChIP-seq experiments showed minimal recruitment (0–6% of nuclei with arrays) (*Figure 4E* and *Table 2A*). In strains with arrays comprised of *Cbr* DCC binding sites, the X chromosomes rarely exhibited fluorescent signal, because the arrays titrated the DCC from X (*Figure 4C*). The titration was so effective that brood sizes of array-bearing hermaphrodites were very low, and hermaphrodite strains carrying arrays could not be maintained. Thus, the twelve high-occupancy *Cbr* DCC binding sites identified by ChIP-seq were named recruitment elements on X (*rex* sites) (*Table 3*), like the *C. elegans* DCC binding sites, due to their ability to recruit the DCC when detached from X.

To determine whether *rex* sites from *C. briggsae* and *C. elegans* had functional overlap in DCC binding specificity, we asked whether a *rex* site from one species could recruit the DCC from the other. We made extrachromosomal arrays in *C. elegans* with DNA from *C. briggsae rex* sites and extrachromosomal arrays in *C. briggsae* with DNA from *C. elegans rex* sites. Arrays in *C. elegans* with *C. briggsae rex* sites failed to recruit the *Cel* DCC or to titrate the *Cel* DCC from *Cel* X chromosomes (*Figure 4C*, *Cbr rex-8*), indicating evolutionary divergence in *rex* sites between the two *Caenorhabditis* species. Reciprocally, extrachromosomal arrays made in *C. briggsae* with *Cel rex* sites failed to bind the *Cbr* DCC or titrate it from the *Cbr* X, confirming divergence in *rex* sites (*Cel rex-33* in *Figure 4D*; *Cel rex-33* and *Cel rex-4* in *Table 2B*). In contrast, controls showed that 100% of extrachromosomal arrays made in *C. elegans* with DNA from either *Cel rex-33* or *Cel rex-4* recruited the *Cel* DCC (*Table 2B*).

Because X chromosomes and extrachromosomal arrays have different topologies, histone modifications, DNA binding proteins, and positions within nuclei, we devised a separate assay to assess the divergence of *rex* sites between species in a more natural chromosomal environment. We inserted six *Cbr rex* sites with a range of ChIP-seq scores into a location on the endogenous *Cel* X chromosome that lacked DCC binding (15, 574, 674 bp) (*Figure 5* and *Table 3*). Proof of principle for the experiment came from finding that insertion of *Cel rex-32*, a high-affinity *Cel* DCC binding site, into the new location on X resulted in DCC binding that was not significantly different from binding at its endogenous location on X (p=0.2, *Figure 5*). All *Cbr rex* sites except *rex-1*, which will be discussed later, failed to recruit the *Cel* DCC when inserted into *Cel* X chromosomes, confirming the divergence of *rex* sites between species.

## Identification of motifs on *Cbr* X chromosomes that recruit the *Cbr* DCC

To understand the mechanisms underlying the selective recruitment of the *Cbr* DCC to X chromosomes, but not autosomes, and the basis for the divergence in X-chromosome targeting between *Caenorhabditis* species, we searched for DNA sequence motifs that are enriched in the twelve *Cbr rex* sites (*Figure 6—figure supplement 1A*) using the website-based program called Multiple Em for Motif Elicitation (MEME) (Version 5.4.1) (*Bailey and Elkan, 1994*; *Bailey et al., 2015*) and compared them to motifs in *C. elegans rex* sites important for recruiting the *Cel* DCC to X (*Figure 6A and B*).

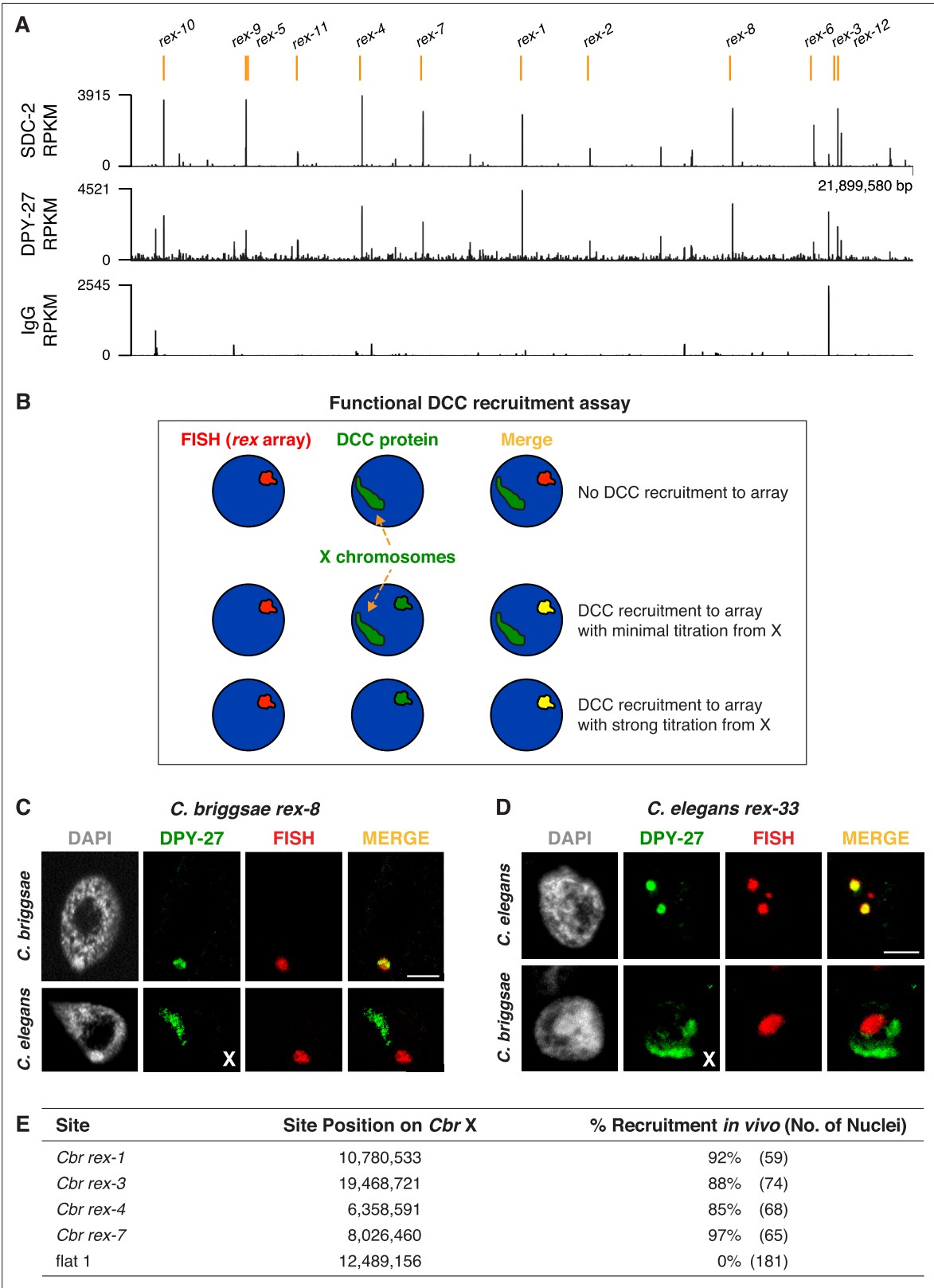

**Figure 4.** Identification of *C. briggsae* dosage compensation complex (DCC) recruitment elements on X. (**A**) ChIP-seq profiles of *Cbr* SDC-2 and *Cbr* DPY-27 binding to X chromosomes. ChIP-seq experiments were performed using an anti-FLAG antibody to immunoprecipitate SDC-2 from a strain encoding FLAG-tagged SDC-2, and the same anti-FLAG antibody was used in ChIP-seq experiments to immunoprecipitate DPY-27 from a strain encoding FLAG-tagged DPY-27. The control IgG ChIP-seq profile on X is also shown. Peaks that correspond to recruitment elements on X (*rex* sites), as

*Figure 4 continued on next page*

*Figure 4 continued*

determined by the assay in (**B**), are indicated in orange above the ChIP-seq profiles. RPKM is the abbreviation for reads per kilobase per million reads mapped. (**B**) Assay performed in vivo to determine whether DNAs from ChIP-seq peaks recruit the DCC when detached from X. XX embryos carrying extrachromosomal arrays with multiple copies of DNA from a ChIP-seq peak in (**A**) were stained with a DNA FISH probe to the array (red) and DPY-27 antibody (green). If the DNA from a peak failed to recruit the DCC, DPY-27 staining would identify X chromosomes but not the array. If DNA from a peak encoded a recruitment site (*rex* site), DPY-27 staining would co-localize with the array and the X chromosome. In the merged image, the array would appear yellow and the X chromosome would appear green. Often, an array carries enough copies of a *rex* site that it titrates most of the DCC from X, and only the array itself shows evidence of DCC binding, appearing yellow in the merged image. In that case, the X chromosome is not detectable by DPY-27 antibody staining. XX strains carrying *rex* arrays that titrate the DCC from X cannot be propagated due to the defect in dosage compensation caused by DCC titration. (**C**) *C. briggsae rex* sites recruit the *C. briggsae* DCC but not the *C. elegans* DCC. Shown is a *C. briggsae* or *C. elegans* XX gut nucleus carrying an extrachromosomal array containing multiple copies of the *C. briggsae* DCC recruitment site *rex-8*. Nuclei were stained with appropriate species-specific *C. briggsae* or *C. elegans* antibodies to the DCC subunit DPY-27 (green), DAPI (gray), and an array FISH probe (red). In *C. briggsae*, DPY-27 bound to arrays in about 40% of the 52 scored nuclei carrying a *Cbr rex-8* array, and the DCC was titrated from X. In *C. elegans*, DPY-27 bound to arrays in 0% of the 27 scored nuclei carrying a *Cbr rex-8* array, and DPY-27 binding to the *C. elegans* X was evident. Scale bar, 5 µm. (**D**) *C. elegans rex* sites do not recruit the *C. briggsae* DCC. Shown is a *C. elegans* or *C. briggsae* XX gut nucleus carrying an extrachromosomal array containing multiple copies of the *C. elegans* recruitment site *rex-33* with three MEX motifs (ln[P] scores of −13.13,−15.33, −15.35). Nuclei were stained with *C. elegans* or *C. briggsae* antibodies to DCC subunit DPY-27 (green), DAPI (gray), and an array FISH probe (red). In *C. elegans*, DPY-27 bound to arrays in 100% of the 63 scored nuclei carrying a *Cel rex-33* array, and the DCC was titrated from X. In *C. briggsae*, DPY-27 bound to arrays in 0% of the 53 scored nuclei carrying a *Cel rex-33* array, but did bind to *Cbr* X chromosomes in the same nuclei (*Table 2*). Scale bar, 5 µm. (**E**) Quantification of exemplary *Cbr* recruitment assays in vivo using extrachromosomal arrays containing multiple copies of DNA from *Cbr* DCC ChIP-seq peaks that define *rex* sites. Data are shown for DPY-27 recruitment to DNA from four strong *Cbr* ChIP-seq peaks and a control region of DNA lacking a DCC peak (flat one containing the gene *mom-1*). Shown are the locations of the sites on X, the total number of embryonic nuclei scored for DPY-27 recruitment to the array, and the percent of nuclei recruiting the DCC. Arrays carrying *rex* sites recruit the DCC but arrays carrying the control flat region fail to recruit the DCC. Results of DCC recruitment assays in vivo for all *rex* sites are presented in *Table 2*.

The online version of this article includes the following figure supplement(s) for figure 4:

**Figure supplement 1.** ChIP-seq profiles of *Cbr* SDC-2 and *Cbr* DPY-27 binding to chromosomes X and V.

We found two motifs enriched within *Cbr rex* sites that are highly enriched on *Cbr* X chromosomes compared to autosomes (*Figure 6A*; *Figure 7A and B*; *Table 3*). A 13 bp motif named MEX (Motif Enriched on X) is enriched up to 12-fold on X chromosomes versus autosomes, and a 30 bp motif named MEX II is enriched up to 30-fold on X versus autosomes (*Figure 7A and B*). All but *rex-11* and *rex-12* had either MEX, MEX II, or both.

The similarity of a motif to the consensus motif is indicated by the ln(P) score, which is the natural log of the probability that the 13-mer for MEX or the 30-mer for MEX II matches the respective consensus motif matrix as calculated by the Patser program (*Hertz and Stormo, 1999*). The lower the score, the better the match. For both MEX and MEX II, the lower the ln(P) score, and hence the better the match to the consensus sequence, the more highly enriched is the motif on X chromosomes compared to autosomes (*Figure 7A and B*).

Our analysis revealed that only the *Cbr* MEX (*Figure 7C*) or MEX II (*Figure 7D*) motifs on X that are located within *rex* sites are bound by SDC-2. Negligible SDC-2 binding was found at single, isolated MEX (*Figure 7C*) or MEX II (*Figure 7D*) motifs on X that are not in *rex* sites. These results implicate MEX and MEX II as important elements for *Cbr* DCC recruitment to *rex* sites.

Neither of the *Cbr* motifs is enriched on the X chromosomes of *C. elegans*, indicating motif divergence between species (*Figure 7A and B*). No additional enriched *C. briggsae* motif candidates were found when the sequences of the two motifs in the twelve *rex* sites were eliminated from the search by converting them to N's and searches for potential motifs were conducted again. In addition, motif analysis of DNA from SDC-2 and DPY-27 ChIP-seq peaks with intermediate or low levels of DCC binding (i.e. lower than for *rex-2*) (*Figure 6—figure supplement 1B*) revealed no motif candidates that correlate with DCC binding.

In *C. elegans*, two motifs are highly enriched on X chromosomes relative to autosomes: a 12 bp motif also called MEX and a 26 bp motif called MEX II (*Figure 6B*; *Fuda et al., 2022*). These *C. elegans* X-enriched motifs are not enriched on *C. briggsae* X chromosomes (*Figure 6B* and *Figure 7—figure supplement 1A, B*). *Cbr* MEX as well as *Cel* MEX and *Cel* MEX II share a common core sequence of CAGGG (*Figure 6*), which is necessary but not sufficient for DCC binding in *C. elegans* (*Fuda et al., 2022*). The core is likely indicative of a common evolutionary history between species. However, the *Cbr* and *Cel* motifs diverged sufficiently that the motifs from one species are not enriched on the

**Table 2.** Results of DCC recruitment assays in vivo.

(A) *Cbr rex* DNA fragments assayed in *C. briggsae* and (B) Identical *Cel rex* DNA fragments assayed in *C. elegans* and in *C. briggsae.*

**(A) *C. briggsae* DCC binds *C. briggsae* DCC recruitment sites.**

| *Cbr rex* Site | *Cbr* Chr X Peak Position | *Cbr* SDC-2 RPKM | *Cbr* Array Assay in vivo % Recruitment (No. of Nuclei) | |
|---|---|---|---|---|
| rex-1 | 10,780,533 | 2890 | 92% | (59) |
| rex-2 | 12,642,866 | 999 | 90% | (101) |
| rex-3 | 19,468,721 | 3219 | 88% | (74) |
| rex-4 | 6,358,591 | 3915 | 85% | (68) |
| rex-5 | 3,153,011 | 3562 | 98% | (45) |
| rex-6 | 18,811,390 | 2203 | 74% | (68) |
| rex-7 | 8,026,460 | 2964 | 97% | (65) |
| rex-8 | 16,578,214 | 3217 | 37% | (52) |
| rex-9 | 3,135,562 | 1029 | 85% | (62) |
| rex-10 | 895,450 | 3605 | 80% | (55) |
| rex-11 | 4,563,250 | 830 | 89% | (54) |
| rex-12 | 19,564,937 | 1786 | 79% | (77) |
| flat 2 | 11,762,995 | 2890 | 6% | (48) |
| flat 3 | 20,918,257 | 999 | 0% | (144) |

**(B) *C. briggsae* DCC does not bind *C. elegans* DCC recruitment sites.**

| *Cel rex* Site | *Cel* Chr X Peak Position | *Cel* Array Assay in vivo % Recruitment (No. of Nuclei) | | *Cbr* Array Assay in vivo % Recruitment (No. of Nuclei) | |
|---|---|---|---|---|---|
| rex-4 | 11,522,205 | 100% | (16) | 1% | (116) |
| rex-33 | 6,296,501 | 100% | (63) | 0% | (53) |

(**A**) Extrachromosomal arrays composed of DNA fragments (2 kb) that were PCR-amplified from *C. briggsae* X chromosome regions corresponding to *Cbr* SDC-2 ChIP-seq peaks were tested for their ability to recruit the *Cbr* DCC. Gut nuclei from *C. briggsae* transgenic lines were scored for the presence of the array using a FISH probe against the *myo-2::gfp* vector and the presence or absence of DCC binding to the array by immunofluorescence signal using *Cbr* DPY-27 antibodies. The % recruitment is the percentage of total scored array-bearing nuclei that showed DPY-27 bound to the array.

(**B**) Identical DNA fragments encoding individual *C. elegans* DCC recruitment sites (*rex*) were injected into *C. elegans* and *C. briggsae* to create extrachromosomal arrays containing multiple copies of the *rex* site. Gut nuclei from *C. elegans* or *C. briggsae* transgenic lines were scored for the presence of the array using a FISH probe against the *myo-2::gfp* vector and for the presence or absence of DCC binding to the array by immunofluorescence signal from the species-matched DPY-27 antibody. The % recruitment is the percentage of total scored array-bearing nuclei that showed DCC binding to the array.

X chromosomes of the other species. Moreover, the *Cbr* MEX motif has a nucleotide substitution that would render the *Cel* MEX motif incapable of binding to the *Cel* DCC. Predominantly, the *C. elegans* consensus MEX motif has a cytosine nucleotide located two nucleotides 5' to the core CAGG G sequence: 5'-TCGCGCAGGGAG-3' (**Figure 6B**). Mutational analysis in *C. elegans* demonstrated that replacing that nucleotide with a guanine greatly reduced DCC binding both in vivo and in vitro (**Fuda et al., 2022**). The consensus *Cbr* MEX motif has a guanine at that critical location, and in principle, the *Cbr* MEX motif would not function as a *Cel* DCC binding motif (**Figure 6**), thereby offering insight into the divergence of X-chromosome binding sites between species.

**Table 3.** Motifs within *rex* sites.

The ln(P) values for MEX II motifs are underlined, and the values for MEX motifs are not underlined.

| *Cbr rex* Site | Chr X Peak Position | SDC-2 RPKM | *Cbr* MEX motif ln(P) < –12 | *Cbr* MEX II ln(P) < –12 |
|---|---|---|---|---|
| *rex-1* | 10,780,533 | 2890 | –15.57 (13 bp) –15.57 (106 bp) –14.63 (14 bp) –14.47 (93 bp) | –27.58 |
| *rex-2* | 12,642,866 | 999 | –14.25 (73 bp) | –22.69 |
| *rex-3* | 19,468,721 | 3219 | –12.36 (178 bp) | –20.04 |
| *rex-4* | 6,358,591 | 3915 | –19.09 (33 bp) –13.80 | |
| *rex-5* | 3,153,011 | 3562 | –18.98 | |
| *rex-6* | 18,811,390 | 2203 | –15.43 (289 bp) –13.35 | |
| *rex-7* | 8,026,460 | 2964 | –18.72 (85 bp) –12.26 (22 bp) –12.58 | |
| *rex-8* | 16,578,214 | 3217 | –13.00 (60 bp) –14.31 (69 bp) –13.22 (23 bp) –13.52 | |
| *rex-9* | 3,135,562 | 1029 | –12.8 | |
| *rex-10* | 895,450 | 3605 | –12.60 (63 bp) –14.68 | |
| *rex-11* | 4,563,250 | 830 | | |
| *rex-12* | 19,564,937 | 1786 | | |

Listed are the *rex* sites analyzed in this study and their motifs. Motif cutoffs used include MEX with ln(P) < –12 and MEX II with ln(P) < –12. The distances between adjacent motifs (in bp) is listed in parenthesis between motifs. Also listed are the coordinates (in bp) with the maximum SDC-2 ChIP-seq signal in each *rex* site and the maximum SDC-2 ChIP signal in reads per kilobase per million reads mapped (RPKM) within a 50 bp window. MEX and MEX II are not likely to be the only DNA sequence features within *rex* sites that contribute to DCC binding, since *rex-11* and *rex-12* lack these motifs with ln(P) values < –12.

In *C. elegans*, a 9 bp motif called Motif C also participates in *Cel* DCC recruitment to X but lacks enrichment on X (*Figure 6B*; *Fuda et al., 2022*). Sequences between the clustered Motif C variants within a *Cel rex* site are also critical for DCC binding (*Fuda et al., 2022*). Evidence that *C. elegans* Motif C fails to participate in *Cbr* DCC recruitment to *Cbr* X chromosomes is our finding that *Cbr* SDC-2 binding is negligible at *Cel* Motif C variants on *Cbr* X, except in the case of rare variants (0.26% of all *Cel* Motif C variants on X) that are within *bona fide* MEX or MEX II motifs in *Cbr rex* sites (*Figure 7—figure supplement 1C*). The likely reason that *Cbr rex-1* recruits the *Cel* DCC when inserted into *Cel* X chromosomes (*Figure 5*) is that each of the four *Cbr* MEX motifs includes a strong match to the consensus *Cel* Motif C (*Figure 5* legend), and DNA sequences surrounding the *Cel* Motif C variants in *Cbr rex-1* are highly conserved with the syntenic region of *C. elegans*, which includes *Cel rex-34*. Both *Cel rex-34* and *Cbr rex-1* are within coding regions of orthologous *pks-1* genes. In contrast, *Cbr rex-7* also contains Motif C variants but lacks the necessary surrounding sequences to permit *Cel* DCC binding when inserted on the *Cel* X (*Figure 5*).

## Mutational analysis of motifs on endogenous *C. briggsae* X chromosomes showed that combinatorial clustering of motifs in *rex* sites facilitates *Cbr* DCC binding but some binding can still occur with only a single motif in a *rex* site

To assess further the importance of the *Cbr* motifs and the divergence of motifs between species, we performed mutational analyses of the two *Cbr* X-enriched motifs. Initial demonstration that both *Cbr* MEX and *Cbr* MEX II motifs participate in DCC binding at *Cbr rex* sites in *C. briggsae* came from analysis using extrachromosomal arrays carrying wild-type and mutant forms of *Cbr rex-1* (*Figure 8—figure supplement 1*). Eighty-nine percent of *C. briggsae* nuclei carrying extrachromosomal arrays composed of wild-type *rex-1* sequences recruited the DCC and titrated it away from X. In contrast, only 24% of nuclei carrying arrays with mutant copies of *rex-1* lacking MEX II recruited the DCC, demonstrating the importance of MEX II. Only 38% of nuclei carrying arrays with mutant copies of *rex-1* lacking all four MEX motifs recruited the DCC, demonstrating the importance of MEX. DCC

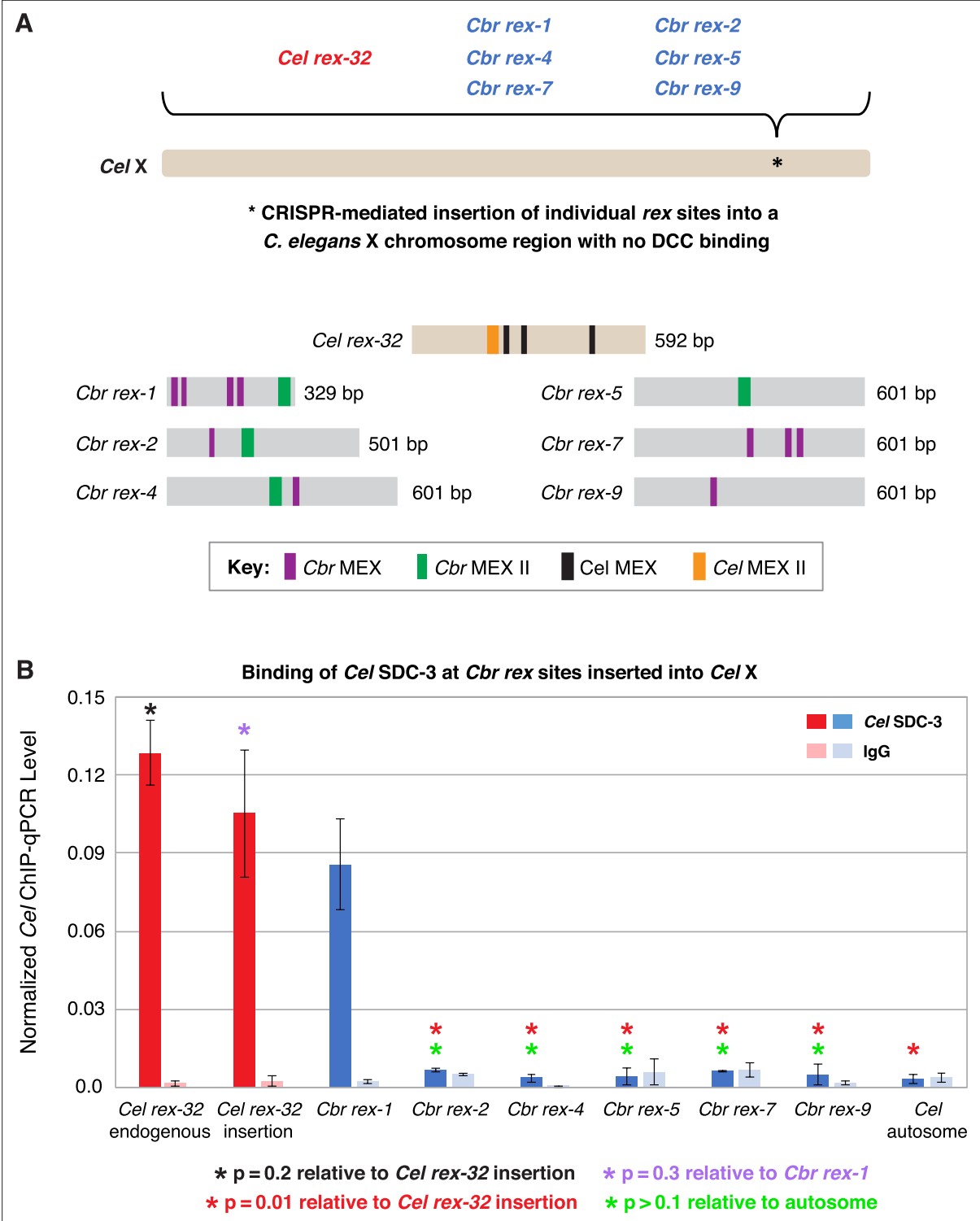

**Figure 5.** *C. briggsae rex* sites integrated into the *C. elegans* X chromosome by genome editing failed to recruit the *C. elegans* dosage compensation complex (DCC). Binding of *C. elegans* DCC protein *Cel* SDC-3 and an IgG control were examined by ChIP-qPCR for *Cel rex-32* at its endogenous location on X, and for six *C. briggsae rex* sites (*Cbr rex-1*, *Cbr rex-2*, *Cbr rex-4*, *Cbr rex-5*, *Cbr rex-7*, and *Cbr rex-9*) plus the control *Cel rex-32* that were inserted by Cas9 genome editing into position 15,574,674 bp of the *C. elegans* X chromosome. (**A**) Schematic shows the location of *Cbr rex* insertions in the *Cel* X chromosomes and shows the different combinations of *Cbr* MEX and MEX II motifs in the inserted *Cbr rex* sites. (**B**) The graph of *Cel* SDC-3 ChIP-qPCR data shows that all *Cbr rex* sites except *rex-1* exhibited SDC-3 binding that was not significantly different from that of the autosomal negative control. *Cbr rex-1* contains a *Cel* Motif C variant within each *Cbr* MEX motif, thereby accounting for the exceptional SDC-3 binding. The Motif

*Figure 5 continued on next page*

*Figure 5 continued*

C variants within *Cbr rex-1* MEX include GGGCAGGGT (–11.68), GGGCAGGGG (–14.16), GCGCAGGGC (–12.06), and CGGCAGGGG (–10.72). A fifth Motif C variant lies between the –14.16 and –12.06 variants: TCCAAGGGG (–9.84). *Cel* SDC-3 levels for each replicate were normalized to the average levels at the five *Cel rex* sites: *Cel rex-8*, *Cel rex-16*, *Cel rex-32*, *Cel rex-48*, and *Cel rex-35*. Error bars represent the SD for three replicates of *Cel rex-32* and *Cbr rex-1* and two replicates for each of *Cbr rex-2*, *Cbr rex-4*, *Cbr rex-5*, *Cbr rex-7*, and *Cbr rex-9*. *Cel* SDC-3 binding to the endogenous *Cel rex-32* site and the inserted *rex-32* site were not significantly different (p=0.2). *Cel* SDC-3 binding to all *Cbr rex* sites except *Cbr rex-1* was significantly lower than binding to the *Cel rex-32* insertion (p=0.01, Student's *t*-test). *Cel* SDC-3 binding at *Cel rex-32* versus *Cbr rex-1* is not significantly different (p=0.3).

binding was reduced to 6% of arrays carrying mutant copies of *rex-1* lacking both MEX II and the four MEX motifs. Hence, both motifs contribute to DCC binding.

This conclusion was reinforced by using genome editing to mutate the MEX II sequence or all MEX II and MEX sequences in the endogenous *rex-1* site on *C. briggsae* X chromosomes and then assaying DCC binding (*Figure 8A–C*). ChIP-seq analysis revealed a significant reduction in DPY-27 binding at *rex-1* lacking MEX II sequences and negligible DPY-27 binding at *rex-1* lacking both MEX and MEX II sequences. Hence, the clustering of motifs in the endogenous *rex-1* on X is important for DCC binding (*Figure 8*).

To evaluate more precisely the participation of different *Cbr* motifs in DCC binding, we used genome editing at three endogenous *rex* sites to evaluate the interplay between MEX and MEX II motifs, only MEX II motifs, or only MEX motifs. Eliminating either MEX or MEX II in *rex-4* reduced binding significantly, but the binding was evident at the remaining motif (*Figure 9A–C* and *Figure 9—figure supplement 1A–C*). Binding was dramatically reduced when both motifs were mutated. This result demonstrates that an individual MEX or MEX II motif can confer significant DCC binding at a *rex* site, but both motifs are needed for full DCC binding.

Equivalent results were found by mutating either of the two MEX II motifs in *rex-3* or combinations of the three MEX motifs in *rex-7*. For *rex-3*, DCC binding was reduced significantly when one of the two MEX II motifs was mutated, but significant binding occurred at either of the remaining MEX II motifs (*Figure 10A–C* and *Figure 10—figure supplement 1A–C*). Binding was greatly reduced when both motifs were mutated. For *rex-7*, DCC binding at the endogenous site lacking the MEX motif with the best match to the consensus sequence (–18.22) was not significantly different from binding at the wild-type site. In contrast, mutating different combinations of two motifs (–18.72 and –12.26 or –18.7 and –12.58) reduced binding significantly (*Figure 11A–C* and *Figure 11—figure supplement 1A–C*). Mutating all three motifs reduced binding severely. Results with the four *Cbr rex* sites, *rex-1*, *rex-3*, *rex-4*, and *rex-7* demonstrate that combinatorial clustering of motifs achieves maximal DCC binding at *Cbr rex* sites, but significant binding can occur at a single motif.

These results contrast with results in *C. elegans*. Mutating individual motifs, either MEX, MEX II, or Motif C, at an endogenous *C. elegans rex* site with multiple different motifs dramatically reduced DCC binding in vivo to nearly the same extent as mutating all motifs, demonstrating synergy in DCC binding (*Fuda et al., 2022*). Hence, not only have the motifs diverged between species, the mode of binding to motifs has also changed.

### Functional divergence of motifs demonstrated by *Cel* DCC binding studies in vivo and in vitro to a *Cel rex* site with *Cbr* MEX and MEX II motifs replacing *Cel* motifs

To explore the divergence in motifs between species in greater detail, we replaced each of the two MEX II motifs of the endogenous *Cel rex-39* site on X with a copy of MEX II from *Cbr rex-3* and assayed the level of *Cel* SDC-3 binding in vivo by ChIP-qPCR (*Figure 12A and B*). SDC-3 binding in vivo was negligible at the *Cel rex-39* site with the *Cbr* MEX II motifs and indistinguishable from binding at the *Cel rex-39* site with two scrambled MEX II motifs, thus demonstrating the high degree of functional divergence between MEX II motifs of different species (*Figure 12B*).

We performed a similar analysis for MEX motifs. We replaced the three MEX motifs in endogenous *Cel rex-33* with the three *Cbr* MEX motifs from endogenous *Cbr rex-7* (*Figure 12D*). SDC-3 binding in vivo was negligible at the *Cel rex-33* site with the *Cbr* MEX motifs and indistinguishable from binding at the *Cel rex-33* site with three scrambled MEX motifs, demonstrating the functional divergence between MEX motifs of different species (*Figure 12E*).

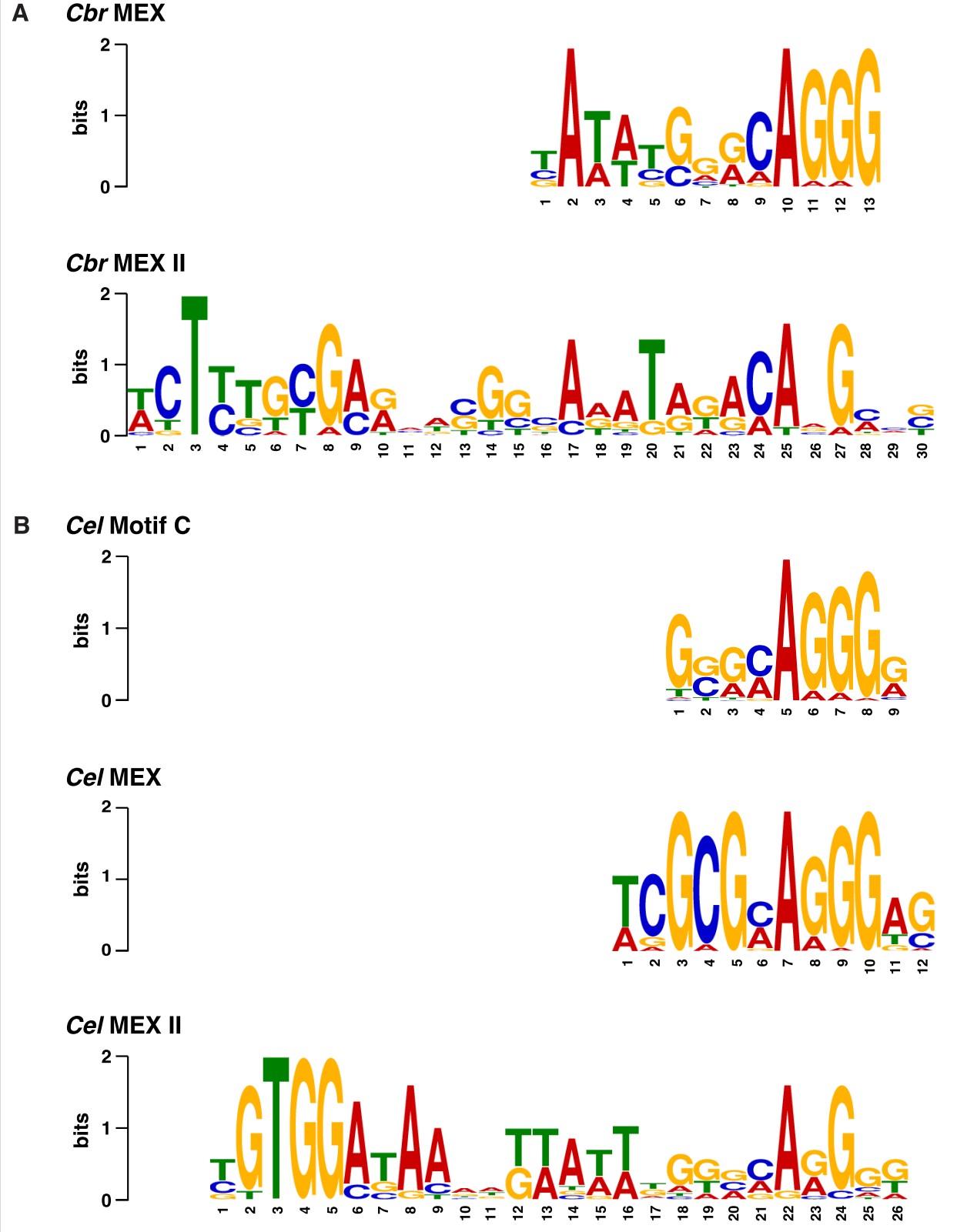

**Figure 6.** Comparison of *C. briggsae* and *C. elegans* DNA motifs on X that occur within respective *rex* sites and recruit respective dosage compensation complex (DCC) complexes. (**A**) Shown are the *C. briggsae* consensus motifs for the 13 bp MEX and 30 bp MEX II variants that recruit the DCC. Also shown are the *C. elegans* consensus motifs for the 12 bp MEX, 26 bp MEX II, and 9 bp Motif C variants that recruit the *Cel* DCC (**B**). The sequences were aligned relative to the conserved adenine in the 5'-CAGGG-3' common core of the motifs. Predominantly, the *Cel* MEX motif has a cytosine in the fourth

*Figure 6 continued on next page*

Figure 6 continued

position of the motif. Mutating it to a guanine (C4G) severely reduced DCC binding in assays conducted in vivo and in vitro. The consensus *Cbr* MEX motif has a guanine at the equivalent position relative to the CAGGG core. Hence, the *Cbr* MEX motif is predicted not to function as a DCC recruitment motif in *C. elegans*.

The online version of this article includes the following figure supplement(s) for figure 6:

**Figure supplement 1.** *C. briggsae* SDC-2 ChIP-seq peak profiles for *rex* sites and non-*rex* sites on X.

As a second approach, we conducted *Cel* DCC binding studies in vitro (Materials and methods). In brief, this assay (*Fuda et al., 2022*) utilized embryo extracts made from a *Cel* nematode strain encoding a 3xFLAG-tagged *Cel* SDC-2 protein expressed from an extrachromosomal array. Wild-type or mutant 651 bp DNA fragments with biotinylated 5' ends were coupled to streptavidin-coated magnetic beads and incubated with embryo extracts. The bound proteins were eluted, spotted onto a nitrocellulose membrane, and probed with a monoclonal mouse anti-FLAG antibody. Antigen-antibody complexes were visualized and quantified by chemiluminescence using an imager. The advantage of this assay is that *Cel* SDC-2 is capable of binding to a single motif on an in vitro template, perhaps because that DNA lacks the competing binding of nucleosomes and general transcription factors that occurs in vivo (*Fuda et al., 2022*).

We assayed *Cel* DCC binding to a *Cel rex-39* site with two *Cbr* MEX II motifs (*Figure 12C*) and to the *Cel rex-33* site with the three *Cbr* MEX motifs (*Figure 12F*). If either of the *Cbr* MEX II motifs inserted into the *Cel rex-39* site were functional or if any of the three *Cbr* MEX motifs inserted into the *Cel rex-33* site were functional, we would detect *Cel* SDC-2 binding to the template in vitro.

The in vitro assay demonstrated robust binding of *Cel* SDC-2 to the wild-type *Cel rex-39* template (*Figure 12C*) and to the wild-type *Cel rex-33* template (*Figure 12F*), as shown previously (*Fuda et al., 2022*). However, *Cel* SDC-2 binding at the *Cel rex-39* site with substituted *Cbr* MEX II motifs was indistinguishable from binding to the mutant *Cel rex-39* template with two scrambled *Cel* MEX II motifs or to the negative control template made from *Cel* X DNA at a site lacking *Cel* DCC binding in vivo (*Figure 12C*). Similarly, *Cel* SDC-2 binding at the *Cel rex-33* site with substituted *Cbr* MEX motifs was indistinguishable from binding to the mutant *Cel rex-33* template with three scrambled *Cel* MEX motifs or to the negative control template (*Figure 12F*). Thus, the in vitro assay demonstrates that substituting *Cbr* MEX II or MEX motifs for *Cel* MEX II or MEX motifs in a *Cel rex* site eliminates *Cel* DCC binding.

## A single nucleotide position in the consensus *Cbr* MEX motif acts as a critical determinant for whether the *Cel* DCC can bind to *Cbr* MEX

In contrast to the many nucleotide changes that mark the difference between MEX II motifs in *C. briggsae* versus *C. elegans*, the MEX motifs are strikingly similar in nucleotide composition and core CAGGG sequence between species (*Figure 6*). A significant change between the consensus MEX motifs is the substitution in *Cbr* MEX of a guanine for the cytosine in *Cel* MEX located two nucleotides 5' from the CAGGG core of both motifs (*Figure 13A*). That C4G transversion was never found in a functional *Cel* MEX motif in vivo or in vitro. Moreover, a C4G change in either the MEX motif of endogenous *Cel rex-1* or in an in vitro Cel DNA template reduced binding to the level of a *rex-1* deletion or negative control lacking a MEX motif (*Fuda et al., 2022*). Hence, the *Cel* DCC would be unable to bind to any *Cbr* MEX motif with C4G. In principle, that single cytosine-to-guanine transversion could be a critical evolutionary change in MEX motifs that render the motifs incapable of binding the DCC from the other species. To test this hypothesis, we made the C4G transversion in each of the three MEX motifs within the endogenous *Cel rex-33* site (*Figure 13B*). *Cel* SDC-3 binding in vivo to the C4G-substituted *Cel rex-33* site was reduced to the same level of binding as that at the *Cel rex-33* site with all three *Cel* MEX motifs scrambled, confirming the functional significance of the nucleotide substitution between species (*Figure 13B*). Our in vitro assay comparing *Cel* SDC-2 binding to the C4G-substituted and the MEX-scrambled *Cel rex-33* DNA templates produced the same negative result (*Figure 13C*).

If the evolutionary transversion of that C to G between *Cel* and *Cbr* MEX motifs represents an important step in the divergence of motif function, then making a G-to-C change within the *Cbr* MEX motifs (G7C) inserted into *Cel rex-33* should enhance *Cel* DCC binding. The substitution would not be

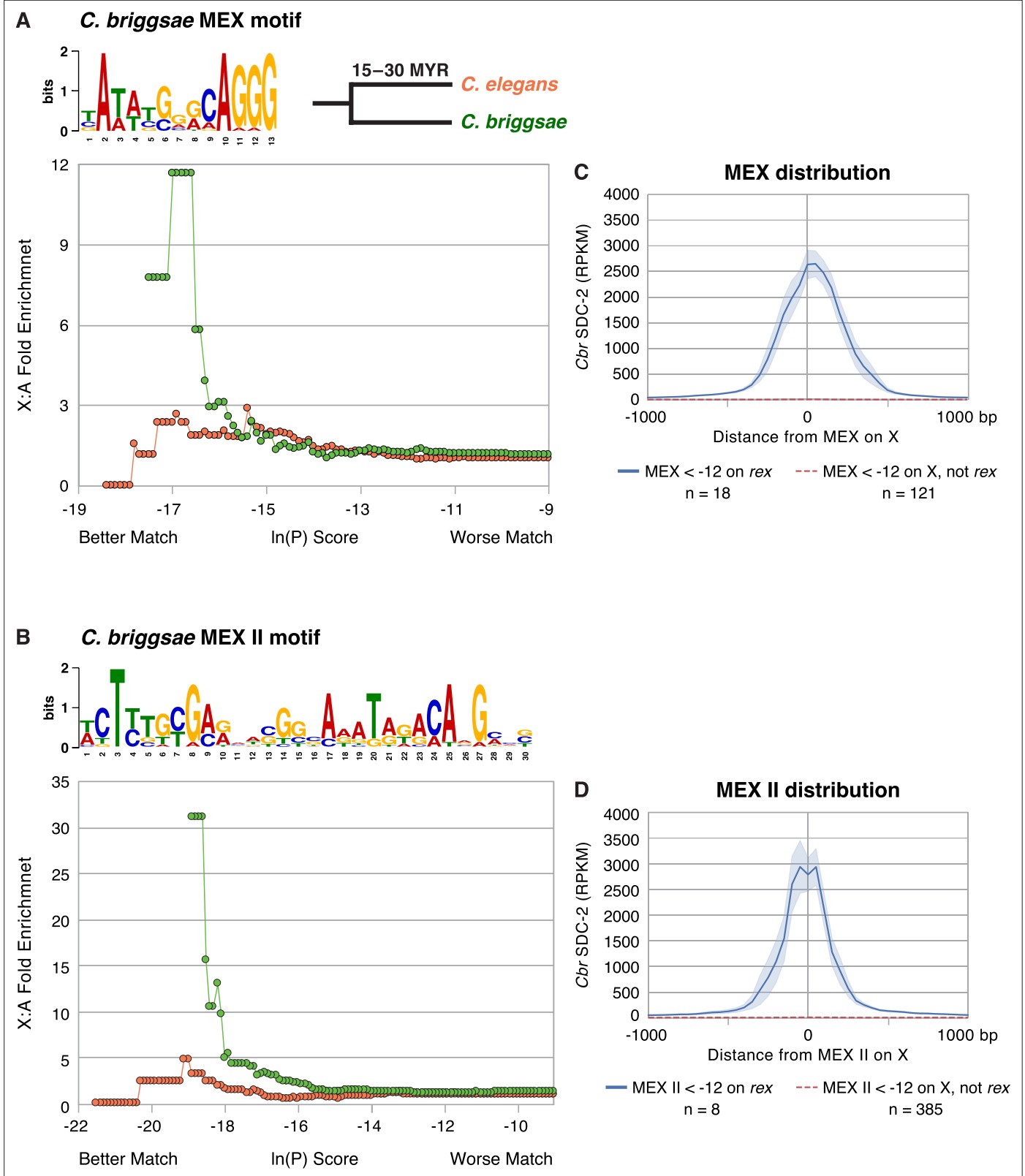

**Figure 7.** Enrichment of *Cbr* MEX and *Cbr* MEX II motifs on X chromosomes between *C. briggsae* and *C. elegans*. (**A, B**) Graphs show the enrichment (y-axis) of *Cbr* MEX (**A**) or *Cbr* MEX II (**B**) variants (x-axis) on X chromosomes compared to autosomes in the *C. briggsae* (green circles) and *C. elegans* (orange circles) genomes. For MEX, the ln(P) is the natural log of the probability that a 13-mer matches the MEX consensus motif matrix (shown above the graphs) as calculated by the Patser program. For MEX II, the ln(P) is the natural log of the probability that a 30-mer matches the MEX II consensus

*Figure 7 continued on next page*

Figure 7 continued

motif matrix (shown above the graphs) as calculated by Patser. The lower the score, the better the match. The maximum theoretical ln(P) value for MEX is –18.7 and for MEX II is –29.3. The best MEX score found on *Cbr* X is –18.7 and for MEX II is - 27.58. The graphs reflect cumulative scores. For example, the 12-fold X:A enrichment of MEX for *C. briggsae* at –17.58 reflects all motifs with ln(P) ≤ –17.58. The *C. elegans* X chromosome is not enriched for the *Cbr* MEX or MEX II consensus motifs that are enriched on *Cbr* X chromosomes and that are pivotal for *Cbr* DCC recruitment to *Cbr* X, as we show subsequently. (**C**) The graph plots the mean (dark blue) and standard error (light blue) of *Cbr* SDC-2 ChIP-seq signal (RPKM) at various distances from MEX motifs (< –12) in *rex* sites versus the mean (dashed red) and standard error (light red) of SDC-2 signal at varying distances from MEX motifs (< –12) on X but not in *rex* sites. Abundant SDC-2 binding was found at MEX motifs in *rex* sites, but negligible SDC-2 binding was found at individual MEX motifs on X that were not in *rex* sites or at MEX motifs on autosomes. n, total number of MEX motifs in each category. (**D**) The graph plots the mean (dark blue) and standard error (light blue) of *Cbr* SDC-2 ChIP-seq signal (RPKM) at various distances from MEX II motifs (< –12) in *rex* sites versus the mean (dashed red line) and standard error (light red) of SDC-2 signal at varying distances from MEX II motifs (< –12) on X but not in *rex* sites. Abundant SDC-2 binding was found at MEX II motifs in *rex* sites, but negligible SDC-2 binding was found at individual MEX II motifs on X that were not in *rex* sites or at MEX II motifs on autosomes. n, total number of MEX II motifs in each category.

The online version of this article includes the following figure supplement(s) for figure 7:

**Figure supplement 1.** The *C. briggsae* X chromosome is not enriched for the *C. elegans* MEX (**A**) or MEX II (**B**) motifs that are highly enriched on *Cel* X chromosomes and pivotal for DCC binding to *Cel* X chromosomes in vivo.

expected to restore *Cel* DCC binding fully, because other sequences within the *Cbr* motif contribute to a lower match to the *Cel* consensus sequence and hence lower *Cel* binding affinity. However, no other identified single nucleotide substitution within a known *Cbr* MEX motif besides C4G is expected to eliminate *Cel* DCC binding (*Fuda et al., 2022*). Indeed, the G7C change to *Cbr* MEX within *Cel rex-33* increased the *Cel* SDC-3 binding in vivo by 4.2-fold and increased the specific *Cel* SDC-2 binding in vitro by 4.3-fold. The G7C change increased *Cel* SDC-3 binding in vivo to 18% of its binding at wild-type *Cel rex-33* (*Figure 13B*) and increased *Cel* SDC-2 binding in vitro to 44% of its the specific binding at the wild-type *Cel rex-33* template (*Figure 13C*). Hence, the cytosine-to-guanine transversion between MEX motifs of *C. elegans* versus *C. briggsae* is important for the functional divergence in motifs.

## Discussion

Comparison of X-chromosome dosage compensation mechanisms between the closely related *Caenorhabditis* species *C. briggsae* and *C. elegans* revealed that both the dosage compensation machinery and the regulatory hierarchy that directs it to hermaphrodite X chromosomes have been conserved, but remarkably, the X-chromosome target specificity of the *C. briggsae* machinery and its mode of binding to X have diverged, as well as the density of DCC recruitment sites. The extent of evolutionary changes in dosage compensation mechanisms between species diverged by only 15–30 MYR is in striking contrast to mechanisms that control somatic sex determination and differentiation in the same species. The master regulator of hermaphrodite sexual fate, TRA-1, is conserved between both species, as is its DNA target specificity (*Berkseth et al., 2013*; *Zarkower and Hodgkin, 1992*). Moreover, the divergence of *Caenorhabditis* dosage compensation mechanisms contrasts with the conservation of *Drosophila* dosage compensation mechanisms (*Alekseyenko et al., 2013*; *Kuzu et al., 2016*) and the conservation of mechanisms controlling developmental processes such as body-plan specification and eye morphogenesis from fruit flies to mice (*Malicki et al., 1990*; *Halder et al., 1995*), which utilize highly conserved transcription factors and *cis*-acting DNA regulatory sequences.

Central to the dosage compensation machinery of both species is a specialized condensin complex. Here we identified two *C. briggsae* dosage compensation proteins (DPY-27 and MIX-1) that are orthologs of the SMC (structural maintenance of chromosome) subunits of condensin and bind to hermaphrodite X chromosomes. As in *C. elegans* (*Chuang et al., 1994*; *Lieb et al., 1998*), mutation of *dpy-27* causes hermaphrodite-specific lethality in *C. briggsae*, and MIX-1 fails to bind X in the absence of DPY-27, consistent with both proteins acting in a complex. We also found that the hermaphrodite-specific *Cbr sdc-2* gene triggers binding of the condensin subunits to X and activates the hermaphrodite mode of sexual differentiation, as in *C. elegans*. Mutation of *Cbr sdc-2* causes XX-specific lethality, and rare XX animals that escape lethality develop as masculinized larvae. SDC-2 and condensin subunits are prevented from binding to the single X of males by the action of *xol-1*, the master sex-determination gene that controls both sex determination and dosage compensation and triggers the male fate by repressing *sdc-2* function. Mutation of *xol-1* kills XO animals because the DCC assembles

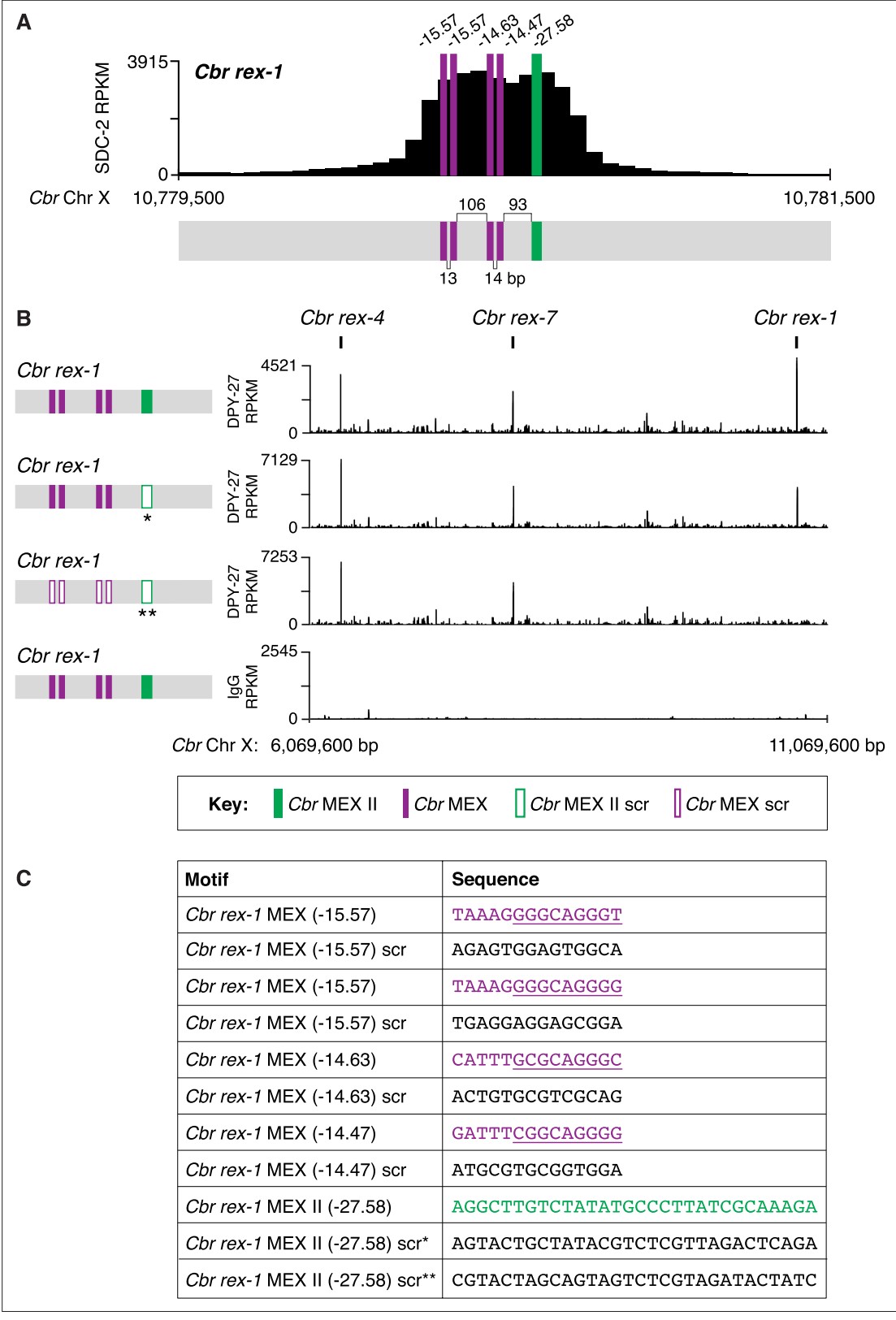

**Figure 8.** Combinatorial clustering of MEX and MEX II motifs in *Cbr rex-1* facilitates dosage compensation complex (DCC) binding to the endogenous *rex-1* site on X. (**A**) Shown is an enlargement of the SDC-2 ChIP-seq peak profile for *Cbr rex-1* with its associated MEX (purple) and MEX II (green) motifs and their ln(P) scores. (**B**) DPY-27 ChIP-seq analysis was performed using anti-FLAG antibody on an otherwise genetically wild-type *C. briggsae* strain encoding FLAG-tagged DPY-27 and on FLAG-tagged DPY-27 *C. briggsae* mutant variants carrying either a scrambled (scr) version of MEX II or a

*Figure 8 continued on next page*

*Figure 8 continued*

scrambled version of MEX II and all four MEX motifs. The control IgG ChIP-seq analysis was performed on the *C. briggsae* strain encoding FLAG-tagged DPY-27 carrying wild-type copies of all *rex* sites. DPY-27 and control IgG ChIP-seq profiles are also shown for *Cbr* sites *rex-7* and *rex-4* as an internal standard since DPY-27 binding was not disrupted at these sites. (**C**) Sequences of the wild-type *Cbr rex-1* MEX motifs and their scrambled versions. Underlined is the *Cel* Motif C variant within each *Cbr* MEX motif. For analyzing MEX II, two different MEX II mutant variants were used, as indicated by asterisks. Numbers between motifs indicate the base pairs separating the motifs. ChIP-seq profiles reveal that mutating only MEX II reduces some DCC binding at *rex-1*, and mutating MEX II and all MEX motifs eliminates DCC binding. The motifs act cumulatively to recruit the DCC.

The online version of this article includes the following figure supplement(s) for figure 8:

**Figure supplement 1.** Clustering of MEX and MEX II motifs in *Cbr rex-1* confers dosage compensation complex (DCC) binding in vivo.

on the single male X, thereby reducing gene expression inappropriately. Mutations in *sdc-2* or *dpy-27* suppress the XO-specific lethality caused by *xol-1* mutations, but only mutations in *sdc-2* permit the rescued animals to develop as males. Just as in *C. elegans*, XO animals rescued by *dpy-27* mutations develop as hermaphrodites, consistent with *dpy-27* controlling only dosage compensation and *sdc-2* controlling both sex determination and dosage compensation. Hence, the two master regulators that control sexual fate and dosage compensation are functionally conserved between the two *Caenorhabditis* species, as is the condensin dosage compensation machinery.

In both species, SDC-2 recruits the condensin DCC subunits to X and is the likely protein to interact directly with X DNA. Despite their central roles in dosage compensation, these 350 kDa proteins lack homology to proteins outside of *Caenorhabditis*, and their only predicted structural feature is a coiled-coil region. Alignment of SDC-2 proteins in five *Caenorhabditis* species revealed only 23–29% identity and 38–45% similarity across the entire protein, with two regions that show greater conservation (*Figure 1—figure supplements 1 and 2A*). One region is N-terminal to the coiled-coil domain and shares 36–45% identity and 57–63% similarity. A second region resides in the C-terminal part of the protein and shows 24–32% identify and 39–51% similarity. Neither region, nor any segment of the protein, has a predicted DNA binding domain. The discovery of any such domain requires ongoing biochemical and structural analysis.

DCC condensin subunits have variable conservation across species, depending on whether they function only in the DCC or participate in other condensin complexes as well. DPY-27, the only condensin subunit specific to the DCC, has only limited conservation: 34% identity and 56% similarity across the *Caenorhabditis* genus (*Figure 1—figure supplement 2B*). In contrast, DCC condensin subunit MIX-1, which also participates in the two condensin complexes required for mitotic and meiotic chromosome segregation, shows greater identity and similarity between both species: 55% and 72%, respectively. In comparison, SMC-4, an ortholog of DPY-27 and a conserved SMC chromosomal ATPase that interacts with MIX-1 in the mitotic and meiotic condensin complexes, but not in the DCC condensin complex (*Hagstrom et al., 2002*; *Csankovszki et al., 2009* and *Mets and Meyer, 2009*), shares even greater conservation between *C. elegans* and *C. briggsae*, commensurate with its universal role in chromosome segregation: 62% identity and 76% similarity. Participation of MIX-1 and SMC-4 in condensin complexes dedicated to chromosome segregation constrains their divergence, thereby accounting for their higher conservation than DPY-27.

Although *C. elegans* and *C. briggsae* have conserved DCC machinery, the DCC binding sites have diverged, as has their density on X. ChIP-seq analysis of *C. briggsae* SDC-2 and DPY-27 revealed twelve sites of binding on X that were validated by functional analysis in vivo as being strong autonomous recruitment (*rex*) sites. Even though the X chromosome of *C. briggsae* (21.5 Mb) is larger than the X of *C. elegans* (17.7 Mb), it has only one-fourth the number of recruitment sites. The *C. briggsae* sites are sufficiently strong that extrachromosomal arrays carrying multiple copies of a single site can titrate the DCC from X and cause dosage-compensation-defective phenotypes in XX animals, including death, as in *C. elegans*. In contrast, extrachromosomal arrays of *C. briggsae rex* sites made in *C. elegans* fail to recruit the *C. elegans* DCC, and *vice versa*, indicating that *rex* sites have diverged between the two species. As a more rigorous test of divergence, individual *C. briggsae rex* sites were inserted in a single copy into *C. elegans* X chromosomes and assayed for *Cel* DCC binding. The *C. elegans* DCC failed to bind to the five *C. briggsae rex* sites inserted into *C. elegans* X chromosomes.

Not only have the *rex* sites diverged, the mechanism by which the Cbr DCC binds to X motifs differs from that of the *Cel* DCC. We identified two motifs within *C. briggsae rex* sites that are highly enriched on X, the 13 bp MEX motif and the 30 bp MEX II motif. Mutating one copy of either motif

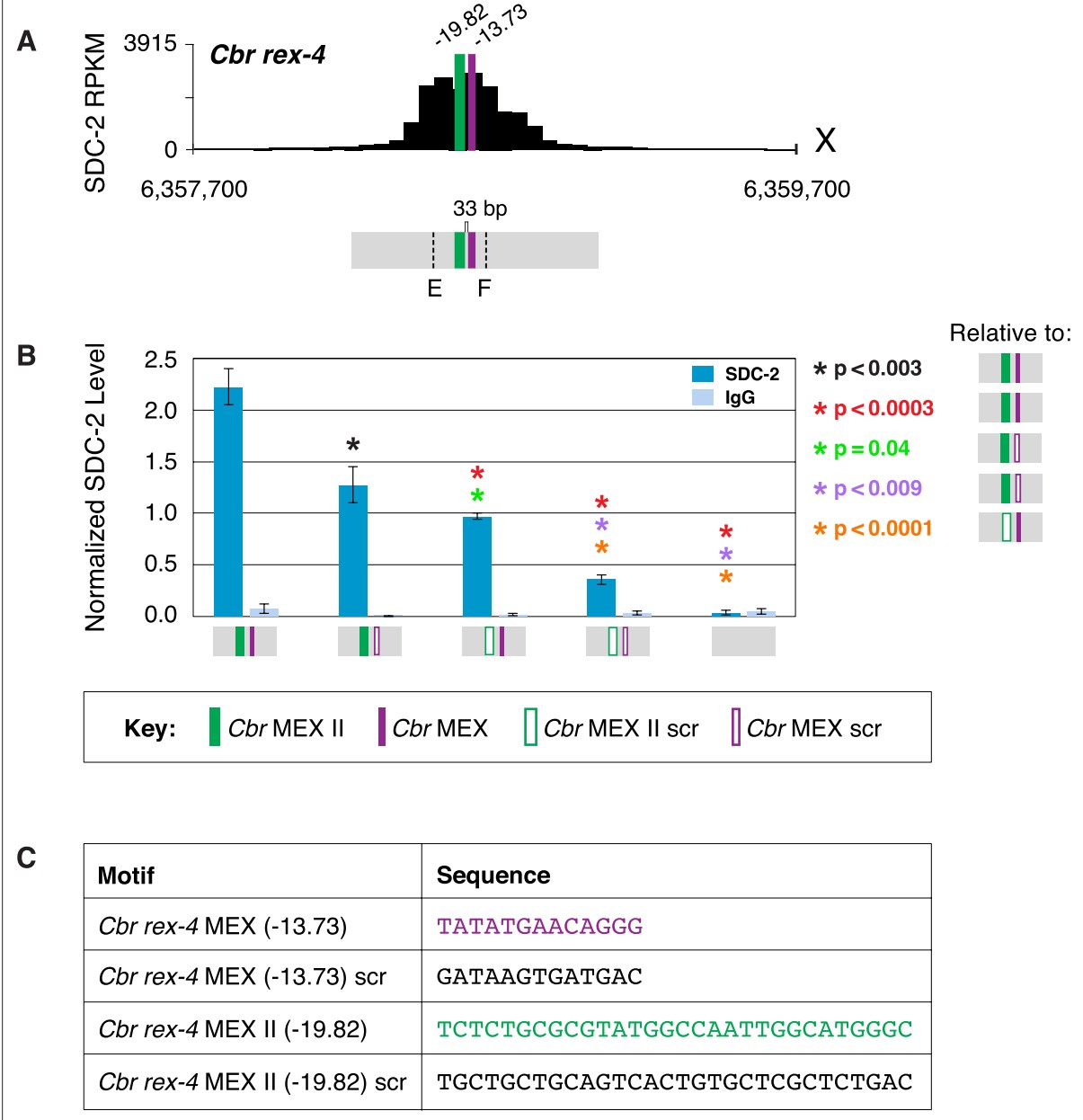

**Figure 9.** MEX and MEX II motifs are critical for dosage compensation complex (DCC) binding to *Cbr rex-4* in vivo. (**A**) Shown is an enlargement of the SDC-2 ChIP-seq profile for *rex-4*, a schematic of the MEX (purple) and MEX II (green) motifs in *rex-4*, and the location of primers (E and F, dashed lines) to evaluate DCC binding in vivo using ChIP-qPCR. Motifs are separated by 33 bp. (**B**) The graph shows ChIP qPCR levels for SDC-2 (dark blue) and control IgG (light blue) at endogenous wild-type *rex-4*, at endogenous *rex-4* with different combinations of motif mutations created by genome editing, and at a negative control site on X of 107 bp that lacks DCC binding centered at (7,000,213 bp). Strains carrying wild-type and mutant motifs encoded FLAG-tagged SDC-2. SDC-2 levels for each replicate were normalized to the average level of five endogenous non-edited *rex* sites (*Cbr rex-1*, *Cbr rex-2*, *Cbr rex-5*, and *Cbr rex-9*). Error bars represent the standard deviation (SD) of three replicates. Asterisks of the same color specify data compared using the Student's *t*-test. If more than two motif combinations are compared, the schematic to the right of the p-value indicates the motif combination to which the other combinations were compared. (**C**) DNA sequences of wild-type and mutant motifs (scr) are shown below the graph. Both MEX and MEX II motifs are critical for DCC binding at *rex-4*. Mutating each motif independently causes an equivalent reduction in DCC binding, and mutating both motifs is necessary to eliminate DCC binding. ChIP-qPCR analysis of SDC-2 binding at intervals across the entire peak is presented in *Figure 9—figure supplement 1*.

The online version of this article includes the following figure supplement(s) for figure 9:

**Figure supplement 1.** MEX and MEX II motifs are critical for SDC-2 binding to *Cbr rex-4* in vivo.

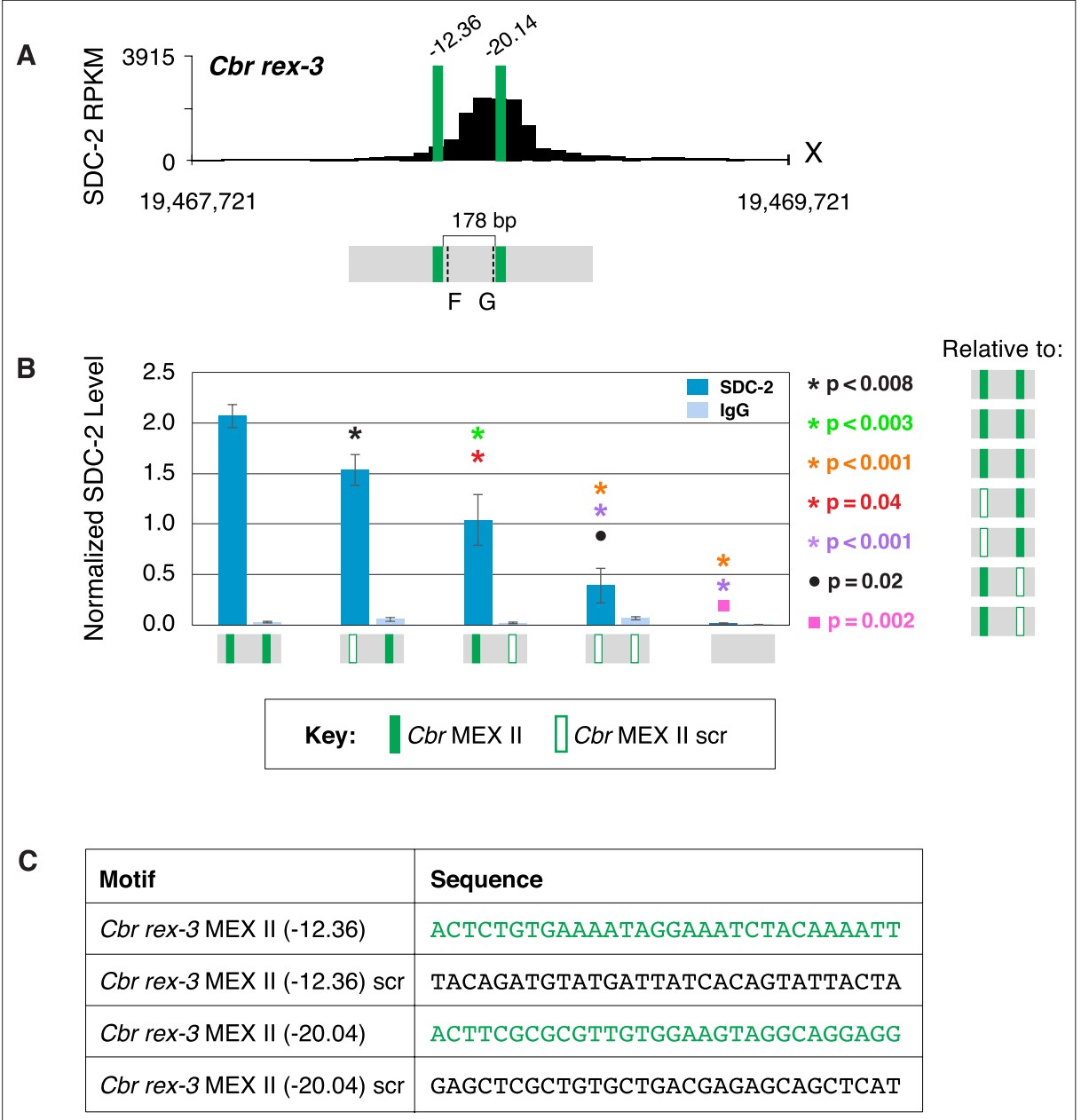

**Figure 10.** Both MEX II motifs are critical for dosage compensation complex (DCC) binding to *Cbr rex-3* in vivo. (**A**) Shown is an enlargement of SDC-2 ChIP-seq profile for *Cbr rex-3* with its associated MEX II motifs (green) and their ln(P) scores. Motifs are separated by 178 bp. Locations of primers (F and G, dashed lines) to evaluate DCC binding in vivo using ChIP-qPCR are shown. (**B**) The graph shows ChIP qPCR levels for SDC-2 (dark blue) and control IgG (light blue) at endogenous wild-type *rex-3*, at endogenous *rex-3* with different combinations of motif mutations created by genome editing, and at a negative control site on X that lacks DCC binding. Strains carrying wild-type and mutant motifs encoded FLAG-tagged SDC-2. SDC-2 levels for each replicate were normalized to the average level of five endogenous non-edited *rex* sites (*Cbr rex-1*, *Cbr rex-2*, *Cbr rex-5*, and *Cbr rex-9*). Error bars represent the standard deviation (SD) of three replicates. Symbols of the same color specify data compared using the Student's *t*-test. If more than two motif combinations are compared, the schematic to the right of the p-value indicates the motif combination to which the other combinations were compared. (**C**) DNA sequences of wild-type and mutant motifs (scr). Both MEX II motifs are critical for DCC binding at *rex-3*. Mutating each motif independently causes an equivalent reduction in DCC binding, and mutating both motifs is necessary to eliminate DCC binding. ChIP-qPCR analysis of SDC-2 binding at intervals across the entire peak is presented in **Figure 10—figure supplement 1**.

The online version of this article includes the following figure supplement(s) for figure 10:

**Figure supplement 1.** Both MEX II motifs are critical for dosage compensation complex (DCC) binding to *Cbr rex-3* in vivo.

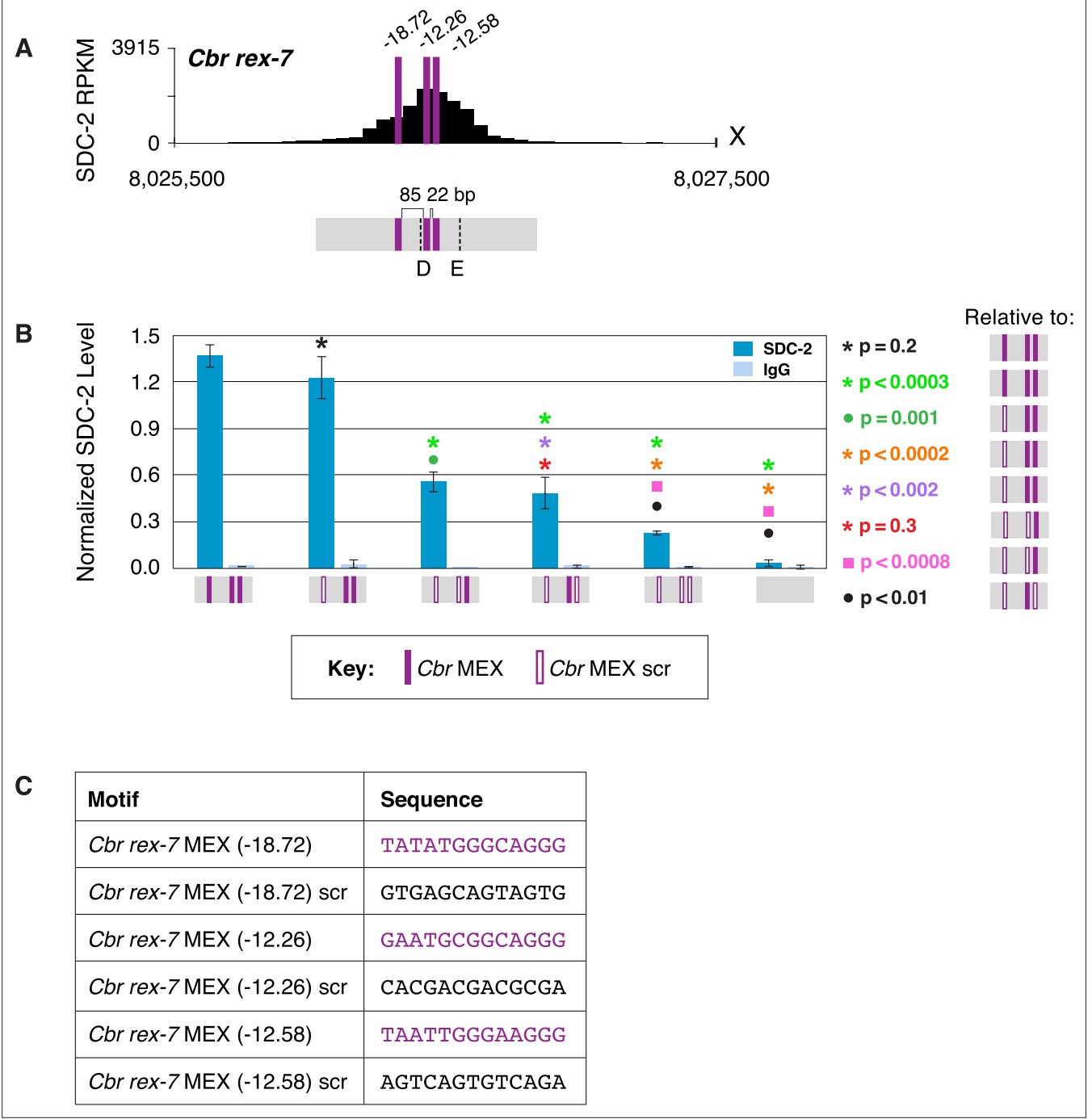

**Figure 11.** Multiple MEX motifs in *Cbr rex-7* contribute to dosage compensation complex (DCC) binding in vivo. (**A**) Shown is an enlargement of SDC-2 ChIP-seq profile for *Cbr rex-7* with its associated MEX motifs (purple) and their ln(P) scores. Motifs are separated by 85 bp and 22 bp. Locations of primers (D and E, dashed lines) to evaluate DCC binding in vivo using ChIP-qPCR are shown. (**B**) The graph shows ChIP qPCR levels for SDC-2 (dark blue) and control IgG (light blue) at endogenous wild-type *rex-7*, at endogenous *rex-7* with different combinations of motif mutations created by genome editing, and at a negative control site on X that lacks DCC binding. Strains carrying wild-type and mutant motifs encoded FLAG-tagged SDC-2. SDC-2 levels for each replicate were normalized to the average level of five endogenous non-edited *rex* sites (*Cbr rex-1*, *Cbr rex-2*, *Cbr rex-5*, and *Cbr rex-9*). Error bars represent the standard deviation (SD) of three replicates. Symbols of the same color specify data compared using the Student's *t*-test. If more than two motif combinations are compared, the schematic to the right of the p-value indicates the motif combination to which the other combinations were compared. (**C**) Sequences of wild-type and mutant motifs (scr). Multiple MEX motifs contribute to DCC binding at *rex-7*. Mutating the first MEX motif has an insignificant effect on DCC binding, but mutating the first MEX motif and either of the other two motifs reduces

*Figure 11 continued on next page*

*Figure 11 continued*

binding equivalently. Mutating all three MEX motifs eliminates DCC binding. ChIP-qPCR analysis of SDC-2 binding at intervals across the entire peak is presented in ***Figure 11—figure supplement 1***.

The online version of this article includes the following figure supplement(s) for figure 11:

**Figure supplement 1.** Multiple MEX motifs in *Cbr rex-7* contribute to dosage compensation complex (DCC) binding in vivo.

in endogenous *rex* sites with multiple motifs reduced binding, but significant binding still occurred at the sites. Binding was eliminated only when all motifs were removed. Hence, DCC binding to motifs in *C. briggsae rex* sites appears additive. In contrast, mutating one motif in *C. elegans rex* sites that have multiple different combinations of motifs reduced binding to nearly the same extent as mutating all motifs, indicating synergy in *C. elegans* DCC binding (***Fuda et al., 2022***).

Additional factors, such as yet-unidentified DNA binding proteins might alter the specificity of DCC binding between species as well as aid DCC binding at *Cbr rex* sites. Precedent exists in the homeodomain family of Hox DNA binding proteins that have remarkably similar DNA specificities for DNA binding in vitro but a wide range of specificities in vivo due to interactions with heterologous cofactors required for functional specificities, such as Pbx-Hox complexes (***Chang et al., 1996***).

The need for synergy in DCC binding to *Cel rex* sites is likely caused by competition between DCC binding and nucleosome formation, since nucleosomes preferentially bind to *rex* sites when DCC binding is precluded by mutations (***Fuda et al., 2022***). The status of nucleosomes on *C. briggsae* X chromosomes remains to be determined. Although a single MEX or MEX II motif enables some DCC binding to a *Cbr rex* site, equivalent motifs on X that are not in *rex* sites appear to lack DCC binding. Nucleosome formation may preclude DCC binding at those motifs. The X may have a paucity of DNA-binding proteins that interact with core histones and open compacted chromatin to enable DCC binding.

Although the X-chromosome motifs of both species share the core consensus sequence CAGGG, the motifs have diverged such that they function in only one species. This functional divergence was demonstrated through DCC binding studies in vivo and in vitro to *C. elegans rex* sites engineered with *C. briggsae* motifs substituted for *C. elegans* motifs. We replaced the two MEX II motifs in the endogenous *C. elegans rex-39* site with *C. briggsae* MEX II motifs and the three MEX motifs in *Cel rex-33* with *Cbr* MEX motifs while maintaining motif spacing appropriate for *C. elegans*. We found negligible *C. elegans* DCC binding in vivo and in vitro. A feature of the in vitro assay is that *Cel* SDC-2 is capable of binding to a single motif on a DNA template, likely because the DNA lacks competing binding of nucleosomes that occurs in vivo. If either *Cbr* MEX II or MEX motif were functional in *C. elegans* we would have detected binding.

While the MEX II motif has diverged sufficiently that evolutionary tracing is difficult, the divergence of MEX motifs provides important insight into their evolution. A major difference in MEX motifs between the two species is the preference for a guanine instead of a cytosine two nucleotides 5' of the conserved CAGGG sequence. We demonstrated that converting that C to G in the three *Cel* MEX motifs of *Cel rex-33* eliminated DCC binding in vitro. Conversely, replacing the G nucleotide in each *Cbr* MEX motif inserted into *Cel rex-33* with a C nucleotide partially restored *Cel* DCC binding in vivo and in vitro, indicating that the single nucleotide change can be important in the evolutionary divergence of this motif. The evolutionary C-to-G substitution in the *Cbr* MEX motif is sufficient to prevent it from functioning in the closely related *C. elegans* species.

In contrast to the divergence of X-chromosome target specificity between *Caenorhabditis* species, X-chromosome target specificity has been conserved among *Drosophila* species. A 21 bp GA-rich sequence motif on X is utilized across *Drosophila* species to recruit the dosage compensation machinery, although it may not be the sole source of X target specificity (***Alekseyenko et al., 2008***; ***Kuzu et al., 2016***; ***Ellison and Bachtrog, 2019***; ***Alekseyenko et al., 2013***).

Conservation of DNA target specificity among species is also a common theme among developmental regulatory proteins that participate in multiple, unrelated developmental processes, such as *Drosophila* Dorsal in the body-plan specification (***Schloop et al., 2020***) or *Caenorhabditis* TRA-1 in hermaphrodite sexual differentiation and male neuronal differentiation (***Berkseth et al., 2013***; ***Bayer et al., 2020***). Typically, for such multi-purpose proteins, target-site specificity is evolutionarily constrained: protein function is changed far more by changes in the number and location of conserved *cis*-acting target sequences than by changes in the target sequences themselves (***Carroll,***

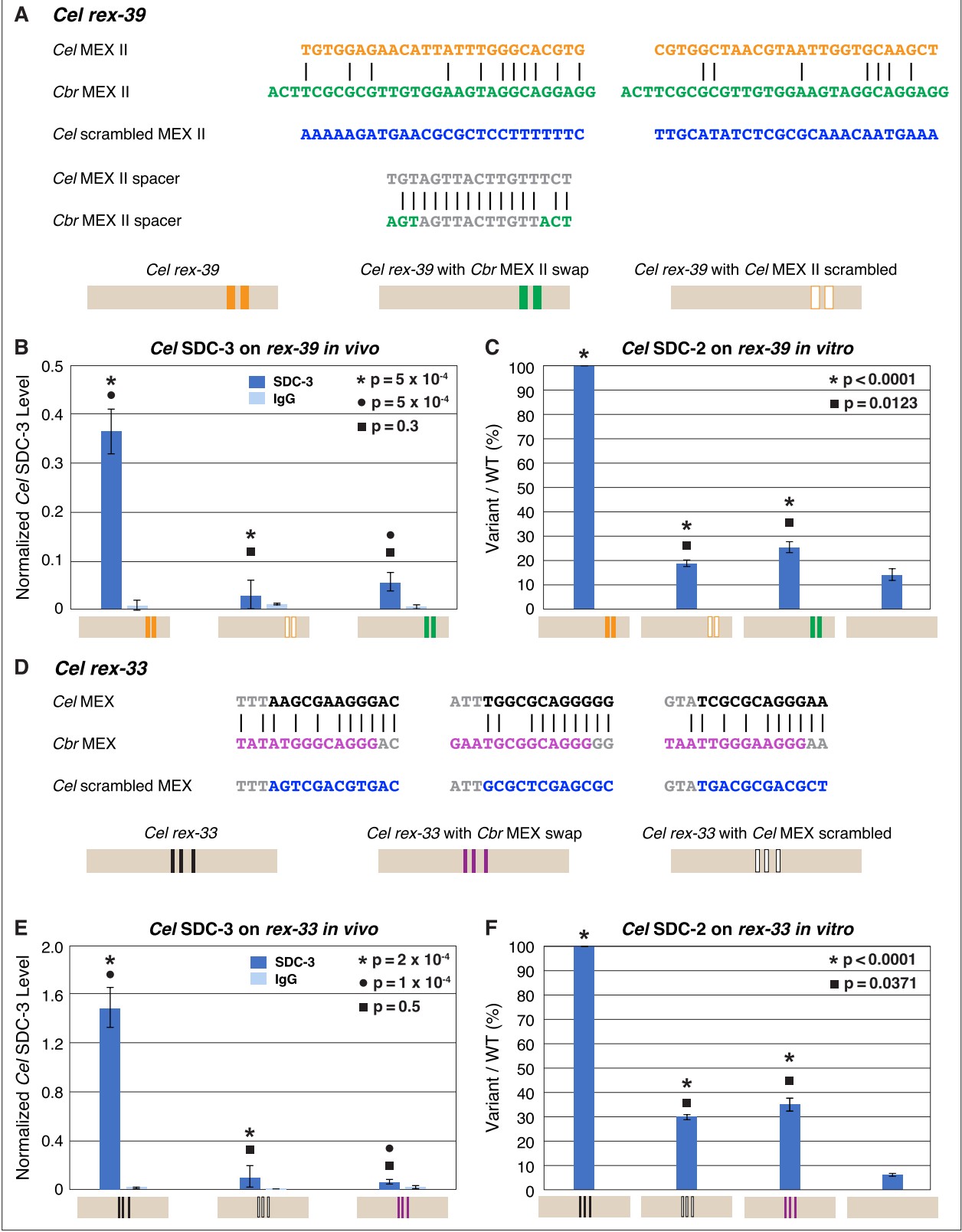

**Figure 12.** Functional divergence of X motifs demonstrated by *C. elegans* dosage compensation complex (DCC) binding studies in vivo and in vitro to *Cel rex* sites engineered to replace *Cel* motifs with *Cbr* MEX and MEX II motifs. (**A**) Comparison of DNA sequences for the two MEX II motifs in wild-type *Cel rex-39* (*Cel* ln[P] of –21.23 and –20.74) with the *Cbr* MEX II motifs (*Cbr* ln[P] of –20.04 and *Cel* ln[P] > –9 for both) that replaced them. DNA sequences of the spacer region between wild-type *Cel* MEX II motifs and inserted *Cbr* MEX II motifs are shown, as are sequences of the scrambled *Cel* MEX II

*Figure 12 continued on next page*

*Figure 12 continued*

motifs used as negative controls. Schematics show keys for *rex* sites analyzed for *Cel* SDC-3 binding in vivo and *Cel* SDC-2 binding in vitro: wild-type *Cel rex-39* (orange, MEX II motifs), *Cel rex-39* with *Cbr* MEX II motifs (green), *Cel rex-39* with scrambled *Cel* MEX II motifs (orange outline). (**B**) Graph shows ChIP qPCR levels for *Cel* SDC-3 (dark blue) and control IgG (light blue) at wild-type *Cel rex-39* and mutant *rex-39* with *Cbr* MEX II motifs in vivo. Cel SDC-3 binds in vivo to endogenous *Cel rex-39* sites with wild-type MEX II motifs but not to mutant *Cel rex-39* sites with either scrambled *Cel* MEX II motifs or *Cbr* MEX II motif replacements. SDC-3 levels for each replicate were normalized to the average SDC-3 level at 7 control *rex* sites (*Cel rex-8, Cel rex-14, Cel rex-16, Cel rex-32, Cel rex-35, Cel rex-36,* and *Cel rex-48*). Error bars represent the standard deviation (SD) of three replicates. Statistical comparisons were calculated using the Student's *t*-test. (**C**) Graph of in vitro assay assessing *Cel* SDC-2 binding to a wild-type *Cel rex-39* DNA template and a mutant *rex-39* template with *Cbr* MEX II motifs. *Cel* SDC-2 binds to the *Cel rex-39* template with wild-type MEX II motifs but not to mutant *rex-39* templates with either scrambled *Cel* MEX II motifs or *Cbr* MEX II motif replacements. *Cel* SDC-2 does not bind to the control template (beige) made of DNA from a site on the *Cel* X that lacks SDC-2 binding in vivo. SDC-2 levels detected for the mutant variants of *rex-39* templates are shown as the percentage (%) of SDC-2 binding to the wild-type *rex-39* template. The plot represents the average of three independent experiments, with error bars indicating SD. Statistical comparisons were calculated using the Student's *t*-test. (**D**) Comparison of DNA sequences for the three MEX motifs in wild-type *Cel rex-33* and the *Cbr* MEX motifs that replaced them. Also shown are sequences for the scrambled *Cel* MEX motifs used as negative controls. Schematics show keys for *rex* sites analyzed for *Cel* SDC-3 binding in vivo and *Cel* SDC-2 binding in vitro: wild-type *Cel rex-33* (black, MEX motifs), *Cel rex-33* with *Cbr* MEX motifs (purple), *Cel rex-39* with scrambled *Cel* MEX motifs (black outline). (**E**) Graph shows ChIP qPCR levels for *Cel* SDC-3 (dark blue) and control IgG (light blue) at wild-type *Cel rex-33* and mutant *rex-33* with *Cbr* MEX motifs in vivo. Cel SDC-3 binds to endogenous *Cel rex-33* sites with wild-type MEX motifs but not to mutant *Cel rex-33* sites with either scrambled *Cel* MEX motifs or *Cbr* MEX motif replacements. Details of the experiment and graph are the same as in (**B**). (**F**) Graph of in vitro assay assessing *Cel* SDC-2 binding to a wild-type *Cel rex-33* DNA template and a mutant *rex-33* template with *Cbr* MEX motifs. *Cel* SDC-2 binds to the *Cel rex-33* template with wild-type MEX motifs but not to mutant *Cel rex-33* templates with either scrambled *Cel* MEX motifs or *Cbr* MEX motif replacements. *Cel* SDC-2 does not bind to the control template (beige). SDC-2 levels detected for the mutant variant *rex-33* templates are shown as the percentage (%) of SDC-2 binding to the wild-type *rex-33* template. The plot represents the average of three independent experiments, with error bars indicating SD. Statistical comparisons were calculated using the Student's *t*-test.

*2008*; *Nitta et al., 2015*). Hence, the divergence in X-chromosome target specificity across the *Caenorhabditis* genus is atypical among developmental regulatory complexes with highly diverse target genes and could have been an important factor for establishing reproductive isolation between species. Our finding is reminiscent of the discovery that centromeric sequences and their corresponding centromere-binding proteins have co-evolved rapidly (*Malik and Henikoff, 2001*; *Henikoff et al., 2001*; *Talbert and Henikoff, 2022*). The occurrence of rapidly changing DNA targets and their corresponding DNA-binding proteins (see also *Liénard et al., 2016*; *Ting et al., 1998*; *Ting et al., 2004*; *Sun et al., 2004*) is an increasingly dominant theme contributing to reproductive isolation.

## Materials and methods

All key resources have been provided in *Supplementary files 1–6*.

### Procedures for mutant isolation

Procedures for *sdc-2* mutant isolation were described previously by *Wood et al., 2011*. *xol-1(y430)*, *dpy-27(y436)*, and *mix-1(y435)* were isolated from a *C. briggsae* deletion library provided by E. Haag using primers listed in *Supplementary file 2*. The resulting strains are listed in *Supplementary file 1*.

### Protein sequence analysis of SDC-2 and DPY-27

Sequence alignments of SDC-2 proteins from *C. elegans* (UniProtKB G5EBL3), *C. brenneri* (UniProtKB G0M6S8), *C. japonica* (WormBase JA61524), *C. tropicalis* (this study), and *C. briggsae* (Uniprot A8XQT3) were generated using Clustal Omega (*Madeira et al., 2022*) and ESPript 3.0 server (https://espript.ibcp.fr) (*Robert and Gouet, 2014*). The coiled-coil annotations were predicted using the web server version of DeepCoil (*Ludwiczak et al., 2019*), part of the MPI Bioinformatics Toolkit (*Zimmermann et al., 2018*; *Gabler et al., 2020*). Pairwise sequence comparisons of SDC-2 proteins were performed with EMBOSS Needle (*Madeira et al., 2022*). Pairwise sequence comparisons of DPY-27 proteins from *C. elegans* (Uniprot P48996), *C. briggsae* (Uniprot A8XX62), *C. brenneri* (WormBase CN00825), *and C. tropicalis* (this study) were generated using EMBOSS Needle.

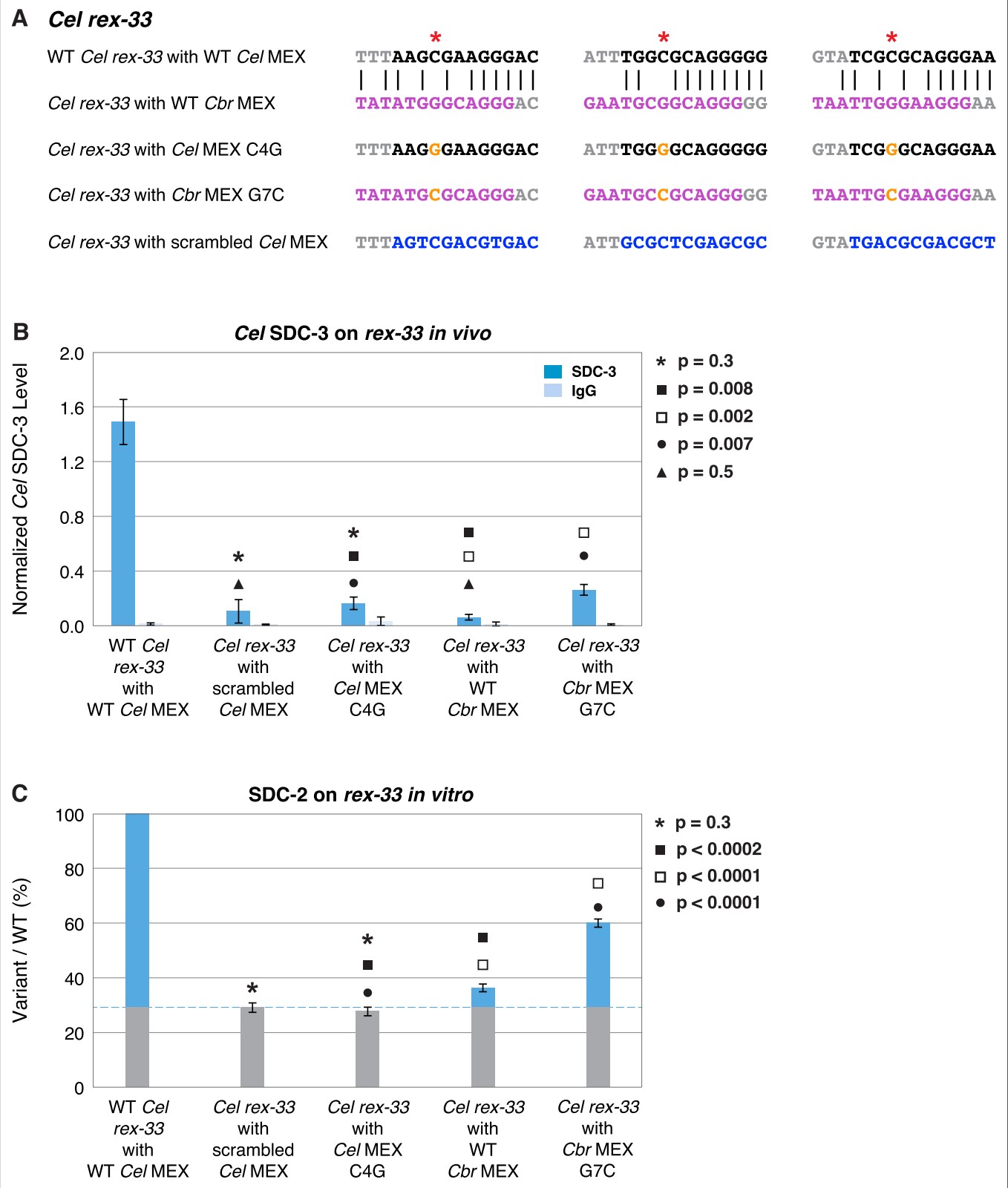

**Figure 13.** A nucleotide position in the consensus *Cbr* MEX motif can act as a critical determinant for whether *Cel* dosage compensation complex (DCC) binds in vivo and in vitro. (**A**) Shown are DNA sequences of three wild-type or mutant *Cel* or *Cbr* MEX motifs within *Cel rex-33* assayed for *Cel* SDC-3 binding in vivo (**B**) and *Cel* SDC-2 binding in vitro (**C**). The ln(P) scores for the wild-type *Cel* MEX motifs in *rex-33* are −13.13, −15.33, and −15.35. The *Cel* ln(P) scores for the 3 substituted *Cbr* MEX motifs are all greater than −9. The three *Cbr* ln[P] scores for those substituted *Cbr* MEX motifs are

*Figure 13 continued on next page*

*Figure 13 continued*

−18.72, −12.26, and −12.58. The *Cel* ln(P) scores for the 3 *Cel* MEX motifs with the C4G change are −9.58, −11.20, and −11.26. The *Cel* ln(P) scores for the three *Cbr* MEX motifs with the G7C change are −12.20, −11.16, and −10.84. The *Cel* ln(P) scores for the *Cel rex-33* scrambled MEX motifs are all greater than −9. (**B**) Graph shows normalized ChIP qPCR levels for C*el* SDC-3 (dark blue) and control IgG (light blue) in vivo at endogenous *Cel rex-33* with wild-type or mutant *Cel* MEX motifs and wild-type or mutant *Cbr* MEX motifs. Replacing the critical cytosine (red asterisk) in each of the three MEX motifs of endogenous *Cel rex-33* with a guanine (C4G) eliminates *Cel* SDC-3 binding, as does scrambling the three C*el* MEX motifs. Substituting three *Cbr* MEX motifs for *Cel* MEX motifs also severely reduces *Cel* DCC binding. Each *Cbr* MEX motif has a guanine instead of a cytosine in the critical location. Replacing the guanine with a cytosine (G7C) in each of the *Cbr* MEX motifs increased C*el* SDC-3 binding 4.2-fold, resulting in a *Cel* SDC-3 binding level representing 18% of that at wild-type *rex-33*. SDC-3 levels for each replicate were normalized to the average SDC-3 level at seven control *rex* sites (*Cel rex-8*, *Cel rex-14*, *Cel rex-16*, *Cel rex-32*, *Cel rex-35*, *Cel rex-36*, and *Cel rex-48*). Error bars represent the standard deviation (SD) of three replicates. Statistical comparisons were calculated using the Student's *t*-test. (**C**) Graph of the in vitro Cel SDC-2 binding assay shows that replacing the critical cytosine (red asterisk) in each of the three MEX motifs of *Cel rex-33* with a guanine (C4G) eliminates *Cel* SDC-2 binding, as does scrambling the three MEX motifs. Substituting three *Cbr* MEX motifs for *Cel* MEX motifs severely reduces *Cel* DCC binding. Each *Cbr* MEX motif has a guanine instead of a cytosine in the critical location. Replacing the guanine with a cytosine (G7C) in each of the *Cbr* MEX motifs increases specific *Cel* SDC-2 binding 4.3-fold and restores it to 44% of that at the wild-type *rex-33* DNA template. SDC-2 levels detected for the mutant variants of *rex-33* templates are shown as the percentage (%) of SDC-2 binding to the wild-type *rex-33* template. The plot represents the average of three independent experiments, with error bars indicating SD. Statistical comparisons were calculated using the Student's *t*-test.

## Preparation of FISH probes

Chromosome FISH probes were prepared from 1 mg of total DNA, which included multiple *C. briggsae* BACs listed in ***Supplementary file 3*** (BACPAC Resources Center, CHORI, Oakland, CA). BACs were purified using the QIAGEN midiprep kit (catalog number 12243). Chromosomal FISH probes were made with the Invitrogen DNA FISH-tag kit. X-chromosome probes (10 BACS covering approximately 5% of the chromosome) were labeled with AlexaFluor 594 (Molecular Probes, F32949), and chromosome III probes (three BACS covering approximately 1% of the chromosome) were labeled with AlexaFluor 488 (Molecular Probes, F32947).

## Preparation of gut nuclei for FISH and immunofluorescence

Adult worms were dissected in 4 µl egg buffer (25 mM HEPES, pH 7.4, 118 mM NaCl, 48 mM KCl, 0.2 mM CaCl$_2$, 0.2 mM MgCl$_2$) on a 18 mm X 18 mm coverslip. 4 µl of 4% formaldehyde (in egg buffer) were added, and the solution was mixed by tapping the coverslip before it was placed onto a Superfrost/Plus glass slide (Fisherbrand, 12-550-15). Fixed samples were incubated for 5 min at room temperature in a humid chamber, then frozen in liquid nitrogen for at least 1 min. Coverslips were removed quickly with a razor blade, and slides were placed immediately into PBS-T (PBS with 1 mM EDTA and 0.5% Triton X-100). Slides were subjected to three 10 min washes in PBS-T at room temperature. Slides were dehydrated in 95% ethanol for 10 min at room temperature followed by either the FISH or immunofluorescence protocol below.

## FISH

Following dehydration of the slides, excess ethanol was removed, 15 µl of hybridization solution (50% formamide, 3 X SSC, 10% dextran sulfate, 10 ng labeled DNA probe in water) was added, and a coverslip was placed on each slide. Slides were placed into a slide chamber, and the FISH incubation protocol was conducted in a PCR machine overnight (80 °C for 10 min, 0.5 °C/sec to 50 °C, 50 °C for 1 min, 0.5 °C/sec to 45 °C, 45 °C for 1 min, 0.5 °C/sec to 40 °C, 40 °C for 1 min, 0.5 °C/sec to 38 °C, 38 °C for 1 min, 0.5 °C/sec to 37 °C, 37 °C overnight). After overnight incubation at 37 °C, slides were washed at 39 °C using the following regime: three times (15 min each) in 2 X SSC (0.3 M NaCl and 30 mM Na$_3$C$_6$H$_5$O$_7$) in 50% formamide, three times (10 min each) in 2 X SSC in 25% formamide, three times (10 min each) in 2 X SSC, and three times (1 min each) in 1 X SSC. Samples were incubated in PBS-T for 10 min at room temperature, and immunofluorescence staining was performed as described below.

## Immunofluorescence of gut nuclei

Following dehydration of slides subjected to immunofluorescence only or to PBS-T treatment (after FISH protocol), the excess liquid was removed (either ethanol from the dehydration step, or PBS-T from FISH protocol) and 20 µl of affinity-purified primary antibodies (*Cbr* DPY-27 and *Cbr* MIX-1 peptide antibodies [Covance, Inc.]) in PBS-T were added at 1:200 dilution. Samples were incubated

in a humid chamber for between 4 hr and overnight. Slides were washed three times (10 min each) in PBS-T at room temperature and then incubated in secondary antibodies for 3–6 hr. Slides were washed three times (10 min each) in PBS-T at room temperature before Prolong (Molecular Probes, P36934) with DAPI (1 µg/ml) was added, and the samples were imaged using a Leica TCS SP2 AOBS. Antibodies used: anti-DPY-27 rabbit antibody raised to *Cbr* DPY-27 C-terminal peptide DVQSEAPS AGRPVETDREGSYTNFD, anti-DPY-27 guinea pig antibody raised to the same *Cbr* DPY-27 peptide, anti-MIX-1 rabbit antibody raised to *Cbr* MIX-1 C-terminal peptide EATKKPSKKSAKKAVQNTDDEME, Alexa Flour 488 goat anti-rabbit antibody (Molecular Probes, A11034), Alexa Flour 488 goat anti-guinea pig antibody (Molecular Probes, A11073), and Alexa Flour 594 goat anti-rabbit antibody (Molecular Probes, A11037).

## Immunofluorescence of embryos

Embryos were picked into 4 µl of water on poly-lysine-treated slides. After adding a coverslip, slides were frozen in liquid nitrogen for at least 1 min. Coverslips were removed rapidly with a razor blade and samples were dehydrated in 95% ethanol for 10 min. Next, 40 µl of fix solution (2% paraformaldehyde in egg buffer) was added and slides were incubated in a humid chamber for 10 min. Slides were washed three times (10 min each) in PBS-T at room temperature. Antibody staining was performed as described above for gut nuclei.

## Immunoprecipitation analysis using DPY-27 antibodies

To determine whether DPY-27, an SMC4 condensin subunit ortholog, interacts with MIX-1, the SMC2 condensin subunit ortholog, we immunoprecipitated proteins with rabbit DPY-27 antibodies and performed mass spectrometry of trypsinized protein bands excised from an SDS-PAGE gel, using protocols from *Mets and Meyer, 2009*. Analysis was performed on proteins within the molecular weight range expected for condensin subunits. In addition to MIX-1 peptides, MALDI-TOF analysis revealed peptides from four common high-molecular weight contaminants in immunoprecipitation experiments (*Table 1*): the three vitellogenin yolk proteins VIT-2, VIT-4, VIT-5, and CBG14234, an ortholog of VIT-4. No protein bands corresponding to the molecular weights of SDC-2 or SDC-3 were visible by SDS-PAGE.

## Western blot analysis of anti-DPY-27 and anti-MIX-1 antibodies

Fifty adult hermaphrodites from strain AF16 [wild-type *C. briggsae*], strain TY5774 *Cbr dpy-27(y706)*, 3xFLAG-tagged *Cbr dpy-27*, or strain TY5005 [*Cbr dpy-27(y436)*] were picked into 25 µL of water, diluted with 25 µL of 2 x SDS Sample Buffer, and heat denatured at 98 °C for 4 min. Samples (20 µL) were fractionated with 3–8% Tris Acetate SDS-PAGE electrophoresis and transferred onto nitrocellulose membranes using standard conditions (60 min at 100 V). Membranes were immunoblotted with either rabbit polyclonal anti-DPY-27 (this study) or rabbit polyclonal anti-MIX antibody (this study). Following incubation with a primary antibody, membranes were incubated with a secondary donkey anti-rabbit HRP antibody (Jackson ImmunoResearch, #711-035-152, RRID: AB_10015282). Nitrocellulose membranes were then incubated in WesternBright Sirius ECL solution (Advansta Corporation, #K-12043-D20) for 2 min, and the chemiluminescence signal was acquired using Image Lab software (Bio-Rad).

## Calculation of viability for *C. briggsae sdc-2* mutants

XX animals: *sdc-2* / + hermaphrodites were crossed to JU935 males, which carry a *gfp* transgene integrated on the X chromosome, and the hermaphrodite cross progeny (*sdc-2* + / + *gfp*) were moved to individual plates. Three classes of genotype were expected among the self-progeny of *sdc-2* + / + *gfp* hermaphrodites. Two classes, (+ *gfp* / + *gfp* and *sdc-2* + / + *gfp*) express GFP, whereas the third class, (*sdc-2* + / *sdc-2* +) does not. If *sdc-2* + / *sdc-2* + animals are 100% viable, the expected proportion of non-green animals among the self-progeny of *sdc-2* + / + *gfp* hermaphrodites is 25%. In each case, the expected number of viable non-green adult progeny is shown in parentheses, and the observed proportion is depicted in the chart as a percentage of the expected number. Wild-type XX viability was calculated among the self-progeny of + + / + *gfp* animals.

XO animals: *sdc-2* + / + *gfp* hermaphrodites were crossed with + + / O (wild-type) males. Successfully mated hermaphrodites were identified by the presence of a copulatory plug and then moved to

individual plates. Two classes of genotypes were expected among the progeny of this cross. One class (+ *gfp* / O) expresses GFP, whereas the other (*sdc-2* + / O) does not. If *sdc-2* + / O animals are 100% viable, the expected proportion of non-green animals among the male progeny is 50%. In each case, the expected number of non-green animals is shown in parentheses, and the observed proportion is depicted in the chart as a percentage of the expected number. Wild-type XO viability was calculated among the male cross-progeny of + + / + *gfp* hermaphrodites and + + / O males.

## Calculation for rescue of *xol-1* XO-specific lethality in *C. briggsae* by an *sdc-2* mutation

The percent viability of wild-type XO animals and mutant XO animals carrying combinations of *xol-1* and *sdc-2* mutations was calculated by formulae that follow. For wild-type XO or *xol-1(y430)* XO progeny from crosses of wild-type or *xol-1(y430)* hermaphrodites mated with wild-type males, the formula is [(number of F1 males)/(total F1 progeny/2)] × 100, a calculation that assumes successful mating and the potential for 50% male cross progeny among the F1. For *xol-1 sdc-2* XO double mutants, *xol-1 sdc-2 / xol-1* hermaphrodites were mated with wild-type males. Given that *xol-1* XO progeny are inviable, *xol-1 sdc-2* F1 males should make up 1/3 of viable F1s. Thus, % XO rescue is calculated as [(number of males)/(total progeny/3)] × 100.

## Genome editing using CRISPR-Cas9

The *Cbr dpy-27(y705)* (*Figure 1F*), *Cbr rex-1* (*Figure 8*), *Cbr rex-3* (*Figure 10*), *Cbr rex-4* (*Figure 9*), and *Cbr rex-7* (*Figure 11*) mutations, as well as *Cel* site 2 insertions (*Figure 5*) and substitutions of *Cbr* MEX motifs into *Cel rex-33* and substitution of *Cbr* MEX II motifs into *Cel rex-39* (*Figure 12*) were made with the CRISPR-Cas9 co-conversion technique using Cas9 RNP injections and species-appropriate co-injection markers (*Farboud et al., 2019*). *C. elegans* editing utilized the *dpy-10* roller marker, and *C. briggsae* editing utilized the *ben-1* marker. The tracrRNA and crRNA guides (Dharmacon) were resuspended in 600 µM of nuclease-free water (Ambion AM9937). The Cas9 RNP mixture for injections included 5 µl Cas9 protein (UC Berkeley QB3 MacroLab, 10 mg/ml), 1.15 µl 2 M HEPES, pH 7.5, 0.35 µl 0.5 M KCl, 0.5 µl 600 µM *dpy-10* crRNA, 1 µl target crRNA (*Supplementary file 4*), 5 µl tracrRNA, and 7 µl nuclease-free water. The Cas9 RNP mix was incubated at 37 °C for 15 min, and 1 µl of the resulting Cas9 RNP mix was combined with 0.5 µl 10 µM *dpy-10* repair oligo (IDT), 0.5 µl 10 µM *rex* repair oligo (IDT), and 8 µl nuclease-free water. After centrifuging at 16,100 × g for 10 min, the Cas9 RNP mix was injected into gonads of adult hermaphrodites. The target-specific sequences for Cas9 guide RNAs are listed in *Supplementary file 4*. The DNA sequences for the repair templates are listed in *Supplementary file 5*.

For *C. elegans*, injected adults were placed on NGM plates. After three days of growth at 25 °C, progeny with the roller phenotype were picked to individual plates and allowed to lay embryos. The roller parents were picked into lysis buffer, and the edited site was amplified and sequenced to identify the worms that were edited. The homozygous progeny from properly edited worms were backcrossed twice to wild-type (N2) worms before usage in experiments. For *C. briggsae*, mutants were isolated as published (*Farboud et al., 2019*). The homozygous progeny from those were backcrossed twice to AF16 worms before usage in experiments. Primers used for genotyping are in *Supplementary file 2*.

## *C. briggsae* ChIP extract preparation

Mixed-stage animals were grown on MYOB agar plates with concentrated HB101 bacteria at 20 °C. Animals were cross-linked with 2% formaldehyde for 10 min and quenched with 100 mM Tris-HCl, pH 7.5. Cross-linked animals were resuspended in 1 ml of FA Buffer (150 mM NaCl, 50 mM HEPES-KOH, pH 7.6, 1 mM EDTA, 1% Triton X-100, 0.1% sodium deoxycholate, 1 mM DTT, and protease inhibitor cocktail [Calbiochem, #539134]) for every 1 gram of animals. This mixture was frozen in liquid nitrogen and then ground under liquid nitrogen by mortar and pestle for 3 min. Once thawed, the mixture was then homogenized with 50 strokes in a Dounce homogenizer. The chromatin was sheared using the Covaris S2 (20% duty factor, power level 8, 200 cycles per burst) for a total of 30 min processing time (60 sec ON, 45 sec OFF, 30 cycles). The concentration of protein in each extract was quantified using the BCA assay (Thermo Fisher, #23228).

## *C. elegans* ChIP-seq extract preparation

Mixed-stage embryos were harvested from hermaphrodites grown on MYOB agar plates with concentrated HB101 bacteria at 20 °C. Embryos were cross-linked with 2% formaldehyde for 10 min and

quenched with 100 mM Tris-HCl, pH 7.5. Cross-linked embryos were resuspended in 1 ml of FA Buffer (150 mM NaCl, 50 mM HEPES-KOH, pH 7.6, 1 mM EDTA, 1% Triton X-100, 0.1% sodium deoxycholate, 1 mM DTT, and protease inhibitor cocktail [Calbiochem, #539134]) for every 1 gram of embryos and homogenized with 50 strokes in a Dounce homogenizer. The chromatin was sheared using the Covaris S2 (20% duty factor, power level 8, 200 cycles per burst) for a total of 30 min processing time (60 sec ON, 45 sec OFF, 30 cycles). The concentration of protein in each extract was quantified using the BCA assay (Thermo Fisher, #23228).

### *C. briggsae* ChIP reactions

To perform the ChIP reactions, a 50 µl bed volume of protein A Dynabeads (Thermo Fisher, #10001D) was re-suspended in 1 ml of FA Buffer (150 mM NaCl, 50 mM HEPES-KOH, pH 7.6, 1 mM EDTA, 1% Triton X-100, 0.1% sodium deoxycholate, 1 mM DTT, and protease inhibitor cocktail [Calbiochem, #539134]). The beads were incubated in a microcentrifuge tube with 5 µg of anti-FLAG antibodies (Sigma-Aldrich, #F1804) and 5 µg of rabbit anti-mouse IgG antibodies (Jackson ImmunoResearch, #315-005-003), or 5 µg of mouse IgG (Sigma-Aldrich, #I5381), and 5 µg of rabbit anti-mouse IgG antibodies (Jackson ImmunoResearch, #315-005-003), for 90 min at room temperature. Tubes with incubated beads were placed on a magnetic rack, and the liquid was discarded.

Extracts containing 2 mg of protein ChIPs were increased in volume to 1 ml with FA buffer and then added to each tube of Dynabeads for a 90 min incubation. The Dynabead-extract mixture was washed at room temperature twice with FA Buffer (150 mM NaCl), once with FA Buffer (1 M NaCl), once with FA Buffer (500 mM NaCl), once with TEL buffer (10 mM Tris-HCl, pH 8.0, 250 mM LiCl, 1% IGEPAL CA-630 [Sigma-Aldrich, #I3021], 1% sodium deoxycholate, 1 mM EDTA), and twice with TE Buffer (10 mM Tris, pH 8.0, 1 mM EDTA). Protein and DNA were eluted with 250 µl of buffer (1% SDS, 250 mM NaCl, 1 mM EDTA) at 65 °C for 20 min.

### *C. elegans* ChIP reactions

To perform the ChIP reactions, a 25 µl bed volume of protein A Dynabeads (Thermo Fisher, #10001D) was re-suspended in 1 ml of FA Buffer (150 mM NaCl, 50 mM HEPES-KOH, pH 7.6, 1 mM EDTA, 1% Triton X-100, 0.1% sodium deoxycholate, 1 mM DTT, and protease inhibitor cocktail [Calbiochem, #539134]). The beads were incubated in a microcentrifuge tube with 3 µg rabbit anti-SDC-3 (lab stock), or 3 µg rabbit IgG (Jackson Immunoresearch, #301-005-003) for 90 min at room temperature. Tubes with incubated beads were placed on a magnetic rack and the liquid was discarded. Protocols for the incubation of extract with beads and elution of protein and DNA from beads were the same as those described for *C. briggsae* ChIP reactions.

### ChIP-seq, illumina sequencing, and data processing

Sequencing libraries were prepared with the eluted materials from ChIP reactions as published (*Zhong et al., 2010*) with minor changes: sequencing adapters were obtained from Bioo (NEXTflex), and adapters were ligated using the NEB Quick Ligation Kit (M2200). Libraries were sequenced on the Illumina HiSeq 4000 platforms. After barcode removal, reads were aligned uniquely to the *C. briggsae* CB4 genome using the default settings in Bowtie version 2.3.4.3. To account for read depth, ChIP signal was normalized to the total number of reads that uniquely aligned to the genome.

### *C. elegans* qPCR

To perform qPCR reactions, protein, and DNA from a *C. elegans* ChIP reaction or from 50% of a control extract (1 mg protein) were de-crosslinked at 65 °C for at least 4 hr with 150 µg/ml Proteinase K (Sigma, #3115887001). DNA from each ChIP reaction or from the control extract was isolated using the Qiagen PCR purification kit and diluted to a final volume of 200 µl with (10 mM Tris-HCl, pH 8.5). For quantitative PCR, the immunoprecipitated DNAs were quantified by comparing their threshold cycle to the standard curve from control DNA (10% and three serial 10-fold dilutions). For the site two insertions, the DCC levels at each inserted *rex* site were calculated for each biological replicate as a ratio of the average DCC level at five control *rex* sites (*rex-8*, *rex-16*, *rex-32*, *rex-35*, and *rex-48*). For all experiments involving endogenous *Cel rex-39* in *Figure 12B* or involving endogenous *Cel rex-33* in *Figure 12E*, the DCC levels at each inserted *rex* site were calculated for each biological replicate as a

ratio of the average DCC level at seven control *rex* sites (*rex-8*, *rex-14*, *rex-16*, *rex-32*, *rex-35*, *rex-36*, and *rex-48*). Primers used for qPCR are listed in *Supplementary file 2*.

## *C. briggsae* qPCR

To perform the qPCR reactions, protein, and DNA from a *C. briggsae* ChIP reaction or from 50% of a control extract (1 mg protein) were de-crosslinked at 65 °C for at least 4 hr with 150 µg/ml Proteinase K (Sigma, #3115887001). DNA from each ChIP reaction or from the control extract was isolated using the Qiagen PCR purification kit and diluted to a final volume of 400 µl with (10 mM Tris-HCl, pH 8.5). For quantitative PCR, the immunoprecipitated DNAs were quantified by comparing their threshold cycle to the standard curve from control DNA (10% and three serial 10-fold dilutions). For the endogenous *rex* site mutations, the DCC levels at each inserted *rex* site were calculated for each biological replicate as a ratio of the average DCC level at four control *rex* sites (*rex-1*, *rex-2*, *rex-5*, and *rex-9*). Primers used for qPCR are listed in *Supplementary file 2*.

## Identification of *C. briggsae* DCC binding motifs

The 500 bp DNA sequence centered on each *C. briggsae* SDC-2 ChIP-seq peak location for the 12 *Cbr rex* sites was isolated from the CB4 reference genome. Motif candidates were obtained by inputting twelve 500 bp sequences onto MEME on the MEME-suite website (*Bailey and Elkan, 1994*; *Bailey et al., 2015*). The settings used to identify motif candidates were the classic mode and any number of repetitions (anr). The X:A enrichment was calculated for motif candidates. The two motif candidates enriched on the *Cbr* X chromosomes were named *Cbr* MEX for the 13 bp motif and *Cbr* MEX II for the 30 bp motif (*Figure 6*).

## X:A motif enrichment calculation

The Patser program (version 3e) (*Hertz and Stormo, 1999*) was used to calculate the natural log of the probability (ln[P]) of finding a match to the *Cbr* MEX motif, *Cbr* MEX II, *Cel* MEX motif, and *Cel* MEX II motif at all positions along each chromosome, as explained in *Fuda et al., 2022*. For each threshold value, the number of motifs with ln[P] values less than the value (better match) was summed for X and for autosomes. The number of autosomal motifs was divided by the total number of autosomal base pairs to find the number of motifs per base pair. The number of motifs per base pair of X was calculated similarly. The final X:A ratio was calculated by dividing the motifs per base pair for X by the motifs per base pair for the autosomes.

## *C. elegans* DCC binding assay performed in vitro

The in vitro *Cel* DCC binding assays (*Figure 12* and *Figure 13*) were performed as described previously in *Fuda et al., 2022*. Briefly, protein extracts for the assays were made from *C. elegans* strain TY4573 [*sdc-2(y74) X; yEx992*], in which the extrachromosomal array *yEx992* carried multiple copies of a transgene that encoded *Cel* SDC-2 tagged with 3xFLAG at its 5' end. Synchronized gravid TY4573 animals were bleached to yield embryos that were resuspended in homogenization buffer (50 mM HEPES, pH 7.5, 140 mM KCl, 1 mM EDTA, 10% v/v glycerol, 0.5% v/v IGEPAL CA-630, 5 mM DTT, 1 mM PMSF, and protease inhibitor cocktail), flash frozen in liquid nitrogen, and stored at –80 °C. After thawing on ice, the embryo suspension was sonicated (Covaris S2) in 1 mL batches for 6 min (duty cycle 10%, intensity five, cycles/burst 200) and centrifuged at 16,100 × *g* for 30 min at 4 °C to pellet embryo debris. The supernatant was removed, flash frozen in liquid nitrogen, and stored at –80 °C. Total protein concentration was determined using the BCA assay (Thermo Fisher, #PI23227).

Both the 601 bp wild-type *rex* DNA and negative control DNA (np1) were obtained by amplifying DNA from worm lysates with oligonucleotides listed in *Supplementary file 6*. Amplified worm DNA was cloned into the TOPOBlunt vector (Thermo Fisher, #450245). The TOPOBlunt-specific oligonucleotides kb157 and kb184r were used to amplify cloned DNA fragments, and the final DNA products had 22 bp (5'-CAGTGTGCTGGAATTCGCCCTT) and 28 bp (5'-GTGATGGATATCTGCAGAATTCGC CCTT-3') sequences added to the 5' and 3' ends, respectively. The TOPOBlunt-specific oligonucleotide kb157 contained a 5' Biotin-TEG moiety (IDTDNA). Mutant and de novo-designed DNA probes were obtained by amplifying gblock fragments using the kb157/kb184r primer pair. The final products assayed in vitro were 651 bp DNA fragments.

For the DNA pull-down assays, Dynabeads M-280 Streptavidin (Thermo Fisher, #11205D) were washed and coupled with the biotinylated DNA (110 ng of DNA per µg of beads) according to manufacturer instructions. After incubation, the beads were washed with buffer A (50 mM Hepes pH 7.5, 70 mM KCl, 10 mM $MgCl_2$, 10% glycerol, 2 mM DTT, 8.5 mg/mL BSA, 1 mM PMSF, and protease inhibitor cocktail), and re-suspended at 15 ng/µL.

Embryo extract was thawed on ice and centrifuged at 16,100 × g for 30 min at 4 °C to remove any aggregates. Embryo extract (800 µg) was incubated with 10 µL of beads (150 ng coupled DNA) in final buffer B (50 mM Hepes pH 7.6, 105 mM KCl, 5 mM $MgCl_2$, 10% v/v glycerol, 0.5 mM EDTA pH 8.0, 0.25% v/v Igepal CA-630, 1 mM DTT, 4.25 mg/mL BSA, 1 mM PMSF, protease inhibitors cocktail, and 1 µg poly(dI-dC)). After a 3–4 hr incubation at 4 °C, samples were centrifuged briefly, incubated on a magnetic rack for 2 min, and the supernatant was removed. Beads were washed with 300 µL buffer B with a short vortex step and placed on ice for 2 min. After incubation, tubes were centrifuged briefly, incubated on a magnetic rack for 2 min, and the supernatant was removed. The wash step was repeated two additional times. For the elution step, magnetic beads were re-suspended in 50 µL buffer C (50 mM Hepes pH 7.5, 2 M NaCl, 5 mM $MgCl_2$, 10% v/v glycerol, 0.5 mM EDTA, 1 mM DTT, and protease inhibitor cocktail) and incubated on ice for 30–45 min. The eluate was transferred to a clean tube, flash frozen in liquid nitrogen, and stored at –80 °C.

For Western dot blot assays, eluted samples (3.5 µL) were spotted in triplicates onto dry nitrocellulose membranes and left to dry for 1 hr at room temperature. Nitrocellulose membranes were incubated in blocking buffer containing milk (5% w/v milk in 1 x TBS supplemented with 0.1% v/v Tween-20) for 1 hr, followed by incubation with primary anti-FLAG antibody (Sigma-Aldrich, #F1804, RRID: AB_262044) for 1 hr, and with secondary donkey anti-mouse HRP antibody (Jackson ImmunoResearch, #715-035-151, RRID: AB_2340771) for 1 hr. Nitrocellulose membranes were incubated in WesternBright Sirius ECL solution (Advansta Corporation, #K-12043-D20) for 3 min, and the chemiluminescence signal was acquired using Image Lab software (Bio-Rad). The dot blot intensities were quantified using the Volume option in Image Lab software.

## Acknowledgements

We are grateful to E Haag and his laboratory for generously providing expertise and reagents to make and screen *C. briggsae* deletion pools, D King for the MALDI-TOF analysis, A Wood for initiating ZFN mutagenesis of *C. briggsae sdc-2*, D Stalford for figure preparation, T Cline and laboratory members for valuable discussions, and the QB3 Genomics Facility (RRID:SCR_022170) for DNA sequencing. This work was supported in part by NIH Grant R35 GM131845 (to BJM). BJM is an investigator of the Howard Hughes Medical Institute.

## Additional information

### Competing interests

Caitlin Schartner: Caitlin Schartner is affiliated with Roche Diagnostics. The author has no financial interests to declare. The other authors declare that no competing interests exist.

### Funding

| Funder | Grant reference number | Author |
| --- | --- | --- |
| Howard Hughes Medical Institute | | Barbara J Meyer |
| National Institutes of Health | R35 GM131845 | Barbara J Meyer |

The funders had no role in study design, data collection and interpretation, or the decision to submit the work for publication.

## Author contributions
Qiming Yang, Katjuša Brejc, Conceptualization, Resources, Data curation, Formal analysis, Validation, Investigation, Visualization, Methodology, Writing – review and editing; Te-Wen Lo, Conceptualization, Resources, Data curation, Formal analysis, Supervision, Validation, Investigation, Visualization, Methodology, Writing – review and editing; Caitlin Schartner, Resources, Formal analysis, Validation, Investigation, Visualization, Methodology; Edward J Ralston, Conceptualization, Resources, Data curation, Formal analysis, Supervision, Validation, Investigation, Visualization, Methodology; Denise M Lapidus, Resources, Data curation, Investigation, Methodology; Barbara J Meyer, Conceptualization, Resources, Data curation, Formal analysis, Supervision, Funding acquisition, Validation, Investigation, Visualization, Methodology, Writing – original draft, Project administration, Writing – review and editing

## Author ORCIDs
Qiming Yang ORCID http://orcid.org/0000-0003-1419-868X
Te-Wen Lo ORCID http://orcid.org/0000-0002-1231-5531
Katjuša Brejc ORCID http://orcid.org/0000-0002-4562-6109
Barbara J Meyer ORCID http://orcid.org/0000-0002-6530-4588

## Decision letter and Author response
Decision letter https://doi.org/10.7554/eLife.85413.sa1
Author response https://doi.org/10.7554/eLife.85413.sa2

# Additional files

## Supplementary files
• MDAR checklist
• Supplementary file 1. List of alleles and strains used in this study.
• Supplementary file 2. List of primers.
• Supplementary file 3. Chromosome-specific BACs used to generate FISH probes.
• Supplementary file 4. List of target-specific sequences for guide RNAs used in CRISPR / Cas9 genome editing experiments.
• Supplementary file 5. DNA sequences of repair templates used in CRISPR / Cas9 genome editing experiments.
• Supplementary file 6. DNA templates used for in vitro DCC binding assays.

## Data availability
GEO GSE214714 is the accession number for the ChIP-seq data reported in this manuscript.

The following dataset was generated:

| Author(s) | Year | Dataset title | Dataset URL | Database and Identifier |
|---|---|---|---|---|
| Yang Q, Lo T, Brejc K, Schartner CM, Ralston EJ, Lapidus DM, Meyer BJ | 2022 | Caenorhabditis briggsae SDC-2 and DPY-27 ChIP-seq | https://www.ncbi.nlm.nih.gov/geo/query/acc.cgi?acc=GSE214714 | NCBI Gene Expression Omnibus, GSE214714 |

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
