## [Editor Report]

This important study uses state-of-the-art methods to explore the evolution of dosage compensation between two closely related nematode species. The evidence supporting the rapid evolution of the recruitment motifs on the X chromosome, despite a general conservation of the dosage compensation machinery, is compelling. This work will be of broad interest to cell biologists and evolutionary biologists.

---

## [Decision Letter]

**Decision letter after peer review:**

Thank you for submitting your article "X-Chromosome Target Specificity Diverged Between Dosage Compensation Mechanisms of Two Closely Related *Caenorhabditis* Species" for consideration by *eLife*. Your article has been reviewed by 3 peer reviewers, one of whom is a member of our Board of Reviewing Editors, and the evaluation has been overseen by a Reviewing Editor and Kevin Struhl as the Senior Editor. The reviewers have opted to remain anonymous.

Essential revisions:

As you will see below, the reviewers bring up a number of points to be addressed. The two main points they agreed on concern the lack of information on the evolution of the proteins themselves (SDC-2 and DCC), as well as a lack of broader context (comparison with other comparative work on DCC and more discussion of the impact of the finding for hybrid incompatibilities and speciation). We expect that you will be able to address these and the other points without too much difficulty.

*Reviewer #1 (Recommendations for the authors):*

1. Antibody validation: the authors raised antibodies against Cbr DPY-27 and MIX-1 that they use for immunostaining. Some additional form of validation of the specificity of the antibodies would be necessary, preferably by western blotting to see the correct protein sizes and their depletion in the respective mutants.

2. Related to the experiments shown in Figures 1 and 2, in which the authors study DCC localization by immunostaining in various mutants, I would recommend the authors be more precise about what is the effect of certain mutations on the "localization" of other DCC components. For example, the authors state that MIX-1 did not localize to X in dpy-27 mutants (line 101), but we don't know whether the lack of signal is due to mislocalization or perhaps MIX-1 is unstable in the absence of DPY-27. Similarly, the explanation of the effects of mutations in xol-1 and SDC-2 could be more precise. The authors use "localization" to describe most of the effects, but in some cases, it seems that the effects of some of the studied mutations are likely to be on expression or stability. If the antibodies raised work well in western blots, it would enable the authors to refine their statements (for SDC-2 they could use their flag-tagged allele). If this is not feasible, the statements in the text should reflect the possible reasons for a lack of signal in the immunostaining.

3. SDC-2 alignments across various Caenorhabditis sp. in comparison with alignments (or % identity and similarity) of other DCC-related proteins. If SDC-2 is the contact with DNA, it may be expected to vary more than other structural components. The similarity between Cbr and Cel SDC-2 proteins seems intuitively low, which could support this idea, but it would be good to know the average conservation between proteins in these two species, and the specific conservation of other proteins in the DCC machinery.

4. DCC composition in Cbr: the authors immunoprecipitated DPY-27 and performed mass spectrometry, but they only focus on the fact that they detected MIX-1 peptides. It would be informative to show the complete mass spectrometry results or at least comment on which other subunits are or are not detected. Is SDC-2 detected?

5. The length of the rex sites used for different experiments is not always obvious, it would be helpful to specify this in the figures and legends. For example, Figure 6, suppl. 1A has no length information on the x-axis.

6. More detailed information on how rex sites were defined seems to be missing. I could not find specific thresholds or criteria to define a rex site. This is important to note (and possibly discuss further) as for example, the two sites defined as rex-11 and -12 do not have any of the motifs that the authors argue are necessary and sufficient to recruit DCC. Also, there is a site between rex-2 and rex-8 that seems like it could have been called a rex site (from Figure 4A), but it is not. It would also be important to show the binding patterns of SDC-2 and DPY-27 on the autosomes in a supplemental figure. There is no mention in the text or figures of how much binding is observed on the other chromosomes.

7. The authors study the binding of DCC to rex sites in different manners. One is an array-based assay in which the authors report whether the array carrying combinations of wt or mutant sequence motifs is able to recruit DPY-27 (as a proxy for DCC binding), shown in Figure 8, suppl 1. In this assay, the authors report the % of nuclei that bind DPY-27 or not and show that losing only the MEX II motif or the MEX motifs, leaving the other motifs intact, reduces the % of nuclei with the signal. But this seems like an unintuitive result for this experiment given their claims that binding to these motifs is additive in Cbr. Shouldn't one expect that most arrays would still recruit DCC, but to a reduced level? In particular given that these arrays contain up to hundreds(?) of copies of the binding site, how do the authors rationalize these observations with their other results?

8. Given that the authors have ChIP-seq data for DCC components in Cel and Cbr, I wonder if they can gain further support for their additive vs. synergistic modes of action by analyzing binding over the whole genome of individual MEX or MEXII motifs.

9. The effects of the single C-G substitution between sites are relatively weak. I would suggest the authors reflect this better in the text, toning down the emphasis on the single nucleotide change in all sections where this is discussed. It is an interesting observation but although this seems to contribute, is clearly not enough to explain the differences in binding between the two species.

10. I would recommend the authors discuss the possible implications of the two different modes of binding. This should have an impact on the specificity of DCC recruitment, and if it doesn't, it would be interesting to think about why not. I guess this is related to points 6 and 7 as well.

*Reviewer #2 (Recommendations for the authors):*

In its current form, however, the manuscript fails to highlight the impact of that discovery.

In the introduction and the conclusion, the authors contrast their findings with the strong conservation of developmental genes (Hox, Wnt, etc.) throughout evolution. This comparison is not really relevant here, and again, the authors should relate more to the work of M. Lynch, H. Malik, S. Henikoff, D. Barbash, etc.

Along the same line, if the rex sites have evolved so fast, one might expect that the proteins of the DCC have also evolved fast. The conservation between the orthologs between the two nematode species appears quite low (compared to other proteins) but this aspect of the comparison is not well documented.

On Line #66 for instance, the authors mention that SDC-2 has "no homologs outside nematodes"..it seems it is also poorly conserved within nematodes. In general, it would be important to know the sequence conservation with other nematodes. Although I do not necessarily request a dN/dS analysis on the evolution of these proteins, having at least an idea of their conservation would be informative.

---

## [Author Response]

Essential revisions:As you will see below, the reviewers bring up a number of points to be addressed. The two main points they agreed on concern the lack of information on the evolution of the proteins themselves (SDC-2 and DCC), as well as a lack of broader context (comparison with other comparative work on DCC and more discussion of the impact of the finding for hybrid incompatibilities and speciation). We expect that you will be able to address these and the other points without too much difficulty.

Thank you. We appreciate the enthusiasm of the reviewers for our work and their insightful comments. As explained below, we have addressed all the concerns of the reviewers in our revised manuscript, including providing more information about the evolution of the proteins, a broader context for our work compared to other dosage compensation mechanisms, and hybrid incompatibilities. We added new suggested data, including Western blots and more images and analysis of the distribution DCC proteins in mutants.

Reviewer #1 (Recommendations for the authors):1. Antibody validation: the authors raised antibodies against Cbr DPY-27 and MIX-1 that they use for immunostaining. Some additional form of validation of the specificity of the antibodies would be necessary, preferably by western blotting to see the correct protein sizes and their depletion in the respective mutants.

In the revised manuscript, we included a new figure (Figure 1-—figure supplement 4A,B) that shows western blots demonstrating antibody specificity for DPY-27 and MIX-1. For DPY-27, the blot shows DPY-27 bands in wild-type adult XX hermaphrodites (lane 1), in adult XX hermaphrodites encoding a 3xFLAG-tagged version of DPY-27 (lane 2), and in mutant adult XX hermaphrodites carrying the *dpy-27(y436)* allele (lane 3).

(A) The DPY-27 antibody recognizes the full-length wild-type DPY-27 protein, a 3xFLAGtagged protein expected to be 3 kDa larger than the wild-type protein, and a mutant protein expected to be 20 kDa shorter than the wild-type protein if translation starts at the internal start codon at position 179, just after the *y436* deletion endpoint. Proteins ran somewhat slower on the PAGE-SDS gels than expected for the calculated molecular weights, but the bands correspond to the appropriate relative molecular weights calculated for the three proteins: 179 kDa for the wild-type protein, 182 kDa for the 3xFLAG-tagged protein, 159 kDa for the mutant protein. The *dpy-27(y436)* mutant lane also shows a faint band at the size expected for the wild-type protein, likely because homozygous *dpy-27(y436)* mutant adults were picked from a plate of *dpy-27(y436)*/+ mothers, and contaminating +/+ or +/*y436* larvae or embryos might have adhered to Dpy adults that were picked.

(B) The MIX-1 western shows a very prominent band of appropriate molecular weight for MIX1 (139 kDa).

2. Related to the experiments shown in Figures 1 and 2, in which the authors study DCC localization by immunostaining in various mutants, I would recommend the authors be more precise about what is the effect of certain mutations on the "localization" of other DCC components. For example, the authors state that MIX-1 did not localize to X in dpy-27 mutants (line 101), but we don't know whether the lack of signal is due to mislocalization or perhaps MIX-1 is unstable in the absence of DPY-27. Similarly, the explanation of the effects of mutations in xol-1 and SDC-2 could be more precise. The authors use "localization" to describe most of the effects, but in some cases, it seems that the effects of some of the studied mutations are likely to be on expression or stability. If the antibodies raised work well in western blots, it would enable the authors to refine their statements (for SDC-2 they could use their flag-tagged allele). If this is not feasible, the statements in the text should reflect the possible reasons for a lack of signal in the immunostaining.

In the revised manuscript, we included new confocal images to clarify more completely the effect of mutations on DPY-27 and MIX-1 binding in vivo (Figure 1D,F). The new Figure 1D shows that DPY-27 exhibits subnuclear localization in *dpy-27*(+) gut cells, consistent with the Xspecific localization demonstrated in panel 1C, and DPY-27 exhibits diffuse nuclear distribution in the rare DPY mutant escapers of the *dpy-27(y436)* strain, as anticipated for a mutant SMC-4 condensin ortholog that lacks most of the N-terminal part of the ATPase domain and, therefore, has no ATP binding or hydrolysis.

The new Figure 1F shows that MIX-1 exhibits subnuclear localization in *dpy-27*(+) gut cells, consistent with the X-specific localization demonstrated in panel 1E. MIX-1 exhibits diffuse nuclear distribution in the *dpy-27(y436)* mutant, like DPY-27 itself, consistent with the participation *Cbr* MIX-1 in a protein complex with *Cbr* DPY-27 and the dependence of MIX-1 on DPY-27 for its binding to X.

All of our *C. briggsae* DPY-27 immunofluorescence experiments demonstrate very low levels of DPY-27 protein in XO embryos and XX embryos lacking *sdc-2.* The residual DPY-27 protein present exhibits diffuse nuclear distribution. This result is identical to the result for *Cel* DPY-27 in *Cel* XO embryos and *Cel sdc-2* mutant XX embryos, where we have shown by western blots that DPY-27 is in very low abundance.

3. SDC-2 alignments across various Caenorhabditis sp. in comparison with alignments (or % identity and similarity) of other DCC-related proteins. If SDC-2 is the contact with DNA, it may be expected to vary more than other structural components. The similarity between Cbr and Cel SDC-2 proteins seems intuitively low, which could support this idea, but it would be good to know the average conservation between proteins in these two species, and the specific conservation of other proteins in the DCC machinery.

In the revised manuscript, we included a new figure that summarizes conservation of SDC-2 (Figure 1-—figure supplement 2A) and DPY-27 (Figure 1-—figure supplement 2B) across the *Caenorhabditis* genus.

For SDC-2, we comment on the degree conservation in both the Results section (page 5, first paragraph) and in the Discussion section (page 19, last paragraph). The summary is as follows.

"Alignment of SDC-2 in five *Caenorhabditis* species revealed only 23-29% identity and 38%45% similarity across the entire protein, with two regions that show greater conservation. One region is N-terminal to the coiled coil domain and shares 36%-45% identity and 57%-63% similarity. A second region resides in the C-terminal part of the protein and shows 24%-32% identify and 39-51% similarity. Neither region, nor any segment of the protein, has a predicted DNA binding domain.

Discovery of any such domain requires ongoing biochemical and structural analysis."

For the DCC condensin subunits we state the following in the Discussion on the bottom page 19 and top of page 20:

"DCC condensin subunits have variable conservation across species, depending on whether they function only in the DCC or participate in other condensin complexes as well. DPY-27, the only condensin subunit specific to the DCC, has only limited conservation: 34% identity and 56% similarity across the *Caenorhabditis* genus (Figure 1—figure supplement 2B). In contrast, DCC condensin subunit MIX-1, which also participates in the two condensin complexes required for mitotic and meiotic chromosome segregation, shows greater identity and similarity between both species: 55% and 72%, respectively. In comparison, SMC-4, an ortholog of DPY-27 and a conserved SMC chromosomal ATPase that interacts with MIX-1 in the mitotic and meiotic condensin complexes, but not in the DCC condensin complex (Hagstrom *et al.,* 2002, Csankovszki *et al.,* 2009 and Mets and Meyer, 2009), shares even greater conservation between *C. elegans* and *C. briggsae*, commensurate with its universal role in chromosome segregation: 62% identity and 76% similarity. Participation of MIX-1 and SMC-4 in condensin complexes dedicated to chromosome segregation constrain their divergence, thereby accounting for their higher conservation than DPY27.”

4. DCC composition in Cbr: the authors immunoprecipitated DPY-27 and performed mass spectrometry, but they only focus on the fact that they detected MIX-1 peptides. It would be informative to show the complete mass spectrometry results or at least comment on which other subunits are or are not detected. Is SDC-2 detected?

In the revised manuscript, we included an expanded section on the immunoprecipitation analysis using DPY-27 antibodies in the Materials and methods (page 25, last paragraph) and also in the legend to Figure 1--table supplement 1. The summary is as follows:

"To determine whether DPY-27, an SMC4 condensin subunit ortholog, interacts with MIX-1, the SMC2 condensin subunit ortholog, we immunoprecipitated proteins with rabbit DPY-27 antibodies and performed mass spectrometry of trypsinized protein bands excised from an SDS-PAGE gel, using protocols from Mets and Meyer (2009). Analysis was performed on proteins within the molecular weight range expected for condensin subunits. In addition to MIX-1 peptides, MALDI-TOF analysis revealed peptides from four common high-molecular weight contaminants in immunoprecipitation experiments (Figure 1--table supplement 1): the three vitellogenin yolk proteins VIT-2, VIT4, VIT-5 and CBG14234, an ortholog of VIT-4. No protein bands corresponding to the molecular weights of SDC-2 or SDC-3 were visible by SDS-PAGE."

5. The length of the rex sites used for different experiments is not always obvious, it would be helpful to specify this in the figures and legends. For example, Figure 6, suppl. 1A has no length information on the x-axis.

In the revised manuscript, the lengths of *rex* sites used for different experiments are explicitly stated in the legends or text for the relevant figures or tables.

6. More detailed information on how rex sites were defined seems to be missing. I could not find specific thresholds or criteria to define a rex site. This is important to note (and possibly discuss further) as for example, the two sites defined as rex-11 and -12 do not have any of the motifs that the authors argue are necessary and sufficient to recruit DCC. Also, there is a site between rex-2 and rex-8 that seems like it could have been called a rex site (from Figure 4A), but it is not. It would also be important to show the binding patterns of SDC-2 and DPY-27 on the autosomes in a supplemental figure. There is no mention in the text or figures of how much binding is observed on the other chromosomes.

The revised manuscript clarifies the discovery and analysis of Cbr *rex* sites (page 10).

"A consistent set of twelve large, overlapping SDC-2 ChIP-seq peaks and DPY-27 ChIP-seq peaks emerged from the studies (Figure 4A), representing less than one-fourth the number of DCC peaks than on the *C. elegans* X chromosome, which is smaller (17.7 Mb for *Cel* vs. 21.5 Mb for *Cbr*). SDC-2 and DPY-27 binding to autosomes was indistinguishable from that of the IgG control (Figure 4—figure supplement 1A,B). To determine whether DNA from these peaks act as autonomous recruitment sites that confer X-chromosome target specificity to the dosage compensation process, we conducted DCC recruitment assays in vivo (Figure 4B). Assays were modeled on *rex* assays developed for *C. elegans* (Materials and methods and Fuda *et al.*, 2022). Embryos carrying extrachromosomal arrays composed of multiple copies of DNA from a single ChIP-seq peak were stained with DPY-27 antibodies and a FISH probe to the array. DPY-27 localized to 80-90% of extrachromosomal arrays carrying DNA from each of the individual peaks (Figure 4C and 4E and Figure 4—table supplement 1A). In contrast, extrachromosomal arrays made from three regions of X lacking DCC binding in ChIP-seq experiments showed minimal recruitment (0-6% of nuclei with arrays) (Figure 4E and Figure 4—table supplement 1A). In strains with arrays comprised of *Cbr* DCC binding sites, the X chromosomes rarely exhibited fluorescent signal, because the arrays titrated the DCC from X (Figure 4C). The titration was so effective that brood sizes of array-bearing hermaphrodites were very low, and hermaphrodite strains carrying arrays could not be maintained. Thus, the twelve high-occupancy *Cbr* DCC binding sites identified by ChIP-seq were named recruitment elements on X (*rex* sites) (Table 1), like the *C. elegans* DCC binding sites, due to their ability to recruit the DCC when detached from X."

Note, in the revised manuscript (Figure 4-—figure supplement 1B), we added the ChIP-seq profiles for both SDC-2 and DPY-27 on chromosome V to emphasize that the autosomal DCC signal is indistinguishable from that of the IgG control on autosomes.

Regarding the usage of motifs to recruit the DCC to rex sites, we state the following in the legend to Table 1, which lists the motifs in *rex* sites:

“MEX and MEX II are not likely to be the only DNA sequence features within res sites that contributing to DCC binding, since rex-11 and rex-12 lack these motifs with ln(P) values < – 12.”

Regarding the small ChIP-seq peak between *rex-2* and *rex-8*, the degree of signal was variable across replicates and was not pursued further.

7. The authors study the binding of DCC to rex sites in different manners. One is an array-based assay in which the authors report whether the array carrying combinations of wt or mutant sequence motifs is able to recruit DPY-27 (as a proxy for DCC binding), shown in Figure 8, suppl 1. In this assay, the authors report the % of nuclei that bind DPY-27 or not and show that losing only the MEX II motif or the MEX motifs, leaving the other motifs intact, reduces the % of nuclei with the signal. But this seems like an unintuitive result for this experiment given their claims that binding to these motifs is additive in Cbr. Shouldn't one expect that most arrays would still recruit DCC, but to a reduced level? In particular given that these arrays contain up to hundreds(?) of copies of the binding site, how do the authors rationalize these observations with their other results?

We do not have any control or knowledge of the copy number of *rex* sites that get included into extrachromosomal arrays. However, if the amount of DCC in a cell is not limiting, we expect to see a difference in the number of nuclei (arrays) recruiting the DCC if we reduce the number of motifs in a *rex* site compared to the number in wild-type sites. We do not see a contradiction in our array vs. ChIP-seq/ChIP-qPCR experiments.

8. Given that the authors have ChIP-seq data for DCC components in Cel and Cbr, I wonder if they can gain further support for their additive vs. synergistic modes of action by analyzing binding over the whole genome of individual MEX or MEXII motifs.

Figure 7 of the original manuscript displayed the X vs. autosome binding profile for SDC2 at MEX (Figure 7C) and MEX II (Figure 7D) motifs. Abundant SDC-2 binding was found at MEX and MEX II motifs in *rex* sites, but negligible SDC-2 binding was found at individual MEX or MEX II motifs on X that were not in *rex* sites or at MEX or MEX II motifs on autosomes. Hence, the additive vs. synergistic modes of DCC binding would need to be explored by other means to gain more understanding.

The Discussion does comment about the difference in binding modes between species (page 21):

“The need for synergy in DCC binding to *Cel rex* sites is likely caused by competition between DCC binding and nucleosome formation, since nucleosomes preferentially bind to *rex* sites when DCC binding is precluded by mutations (Fuda *et al.*, 2022). The status of nucleosomes on *C. briggsae* X chromosomes remains to be determined. Although a single MEX or MEX II motif enables some DCC binding to a *Cbr rex* site, equivalent motifs on X that are not in *rex* sites appear to lack DCC binding. Nucleosome formation may preclude DCC binding at those motifs. The X may have a paucity of DNA-binding proteins that interact with core histones and open compacted chromatin to enable DCC binding.”

9. The effects of the single C-G substitution between sites are relatively weak. I would suggest the authors reflect this better in the text, toning down the emphasis on the single nucleotide change in all sections where this is discussed. It is an interesting observation but although this seems to contribute, is clearly not enough to explain the differences in binding between the two species.

In the Results (pages 17-18), we explain that the C4G substitution is important and why:

"In contrast to the many nucleotide changes that mark the difference between MEX II motifs in *C. briggsae* versus *C. elegans*, the MEX motifs are strikingly similar in nucleotide composition and core CAGGG sequence between species (Figure 6). A significant change between the consensus MEX motifs is the substitution in *Cbr* MEX of a guanine for the cytosine in *Cel* MEX located two nucleotides 5' from the CAGGG core of both motifs (Figure 13A). That C4G transversion was never found in a functional *Cel* MEX motif in vivo or in vitro*.* Morevoer, a C4G change in either the MEX motif of endogenous *Cel rex-1* or in an in vitro *Cel* DNA template reduced binding to the level of of a *rex-1* deletion or negative control lacking a MEX motif (Fuda *et al.*, 2022). Hence, the *Cel* DCC would be unable to bind to any *Cbr* MEX motif with C4G. In principle, that single cytosine-to-guanine transversion could be a critical evolutionary change in MEX motifs that render the motifs incapable of binding the DCC from the other species. To test this hypothesis, we made the C4G transversion in each of the three MEX motifs within the endogenous *Cel rex-33* site (Figure 13B). *Cel* SDC-3 binding in vivo to the C4G-substituted *Cel rex-33* site was reduced to the same level of binding as that at the *Cel rex-33* site with all three *Cel* MEX motifs scrambled, confirming the functional significance of the nucleotide substitution between species (Figure 13B). Our in vitro assay comparing *Cel* SDC-2 binding to the C4G-substituted and the MEX-scrambled *Cel rex-33* DNA templates produced the same negative result (Figure 13C).

If the evolutionary transversion of that C to G between *Cel* and *Cbr* MEX motifs represents an important step in the divergence of motif function, then making a G-to-C change within the *Cbr* MEX motifs (G7C) inserted into *Cel rex-33* should enhance *Cel* DCC binding. The substitution would not be expected to restore *Cel* DCC binding fully, because other sequences within the *Cbr* motif contribute to a lower match to the *Cel* consensus sequence and hence lower *Cel* binding affinity. However, no other identified single nucleotide substitution within a known *Cbr* MEX motif besides C4G is expected to eliminate *Cel* DCC binding (Fuda *et al.,* 2022). Indeed, the G7C change to *Cbr* MEX within *Cel rex-33* increased the *Cel* SDC-3 binding in vivo by 4.2-fold and increased the specific *Cel* SDC-2 binding in vitro by 4.3-fold. The G7C change increased *Cel* SDC-3 binding in vivo to 18% of its binding at wild-type *Cel rex-33* (Figure 13B) and increased *Cel* SDC-2 binding in vitro to 44% of its the specific binding at the wild-type *Cel rex-33* template (Figure 13C)*.* Hence, the cytosine-toguanine transversion between MEX motifs of *C. elegans* versus *C. briggsae* is important for the functional divergence in motifs."

In the Discussion (page 22) we explain the following:

"A major difference in MEX motifs between the two species is the preference for a guanine instead of a cytosine two nucleotides 5' of the conserved CAGGG sequence. We demonstrated that converting that C to G in the three *Cel* MEX motifs of *Cel rex-33* eliminated DCC binding in vitro. Conversely, replacing the G nucleotide in each *Cbr* MEX motif inserted into *Cel rex-33* with a C nucleotide partially restored *Cel* DCC binding in vivo and in vitro, indicating that the single nucleotide change can be important in the evolutionary divergence of this motif. The evolutionary C-to-G substitution in the *Cbr* MEX motif is sufficient to prevent it from functioning in the closely related *C. elegans* species."

10. I would recommend the authors discuss the possible implications of the two different modes of binding. This should have an impact on the specificity of DCC recruitment, and if it doesn't, it would be interesting to think about why not. I guess this is related to points 6 and 7 as well.

Depending on the conditions, we would not necessarily expect to see an impact on the specificity of DCC recruitment in the two different modes of binding. Without a different set of experiments, we cannot address this question.

Reviewer #2 (Recommendations for the authors):In its current form, however, the manuscript fails to highlight the impact of that discovery.In the introduction and the conclusion, the authors contrast their findings with the strong conservation of developmental genes (Hox, Wnt, etc.) throughout evolution. This comparison is not really relevant here, and again, the authors should relate more to the work of M. Lynch, H. Malik, S. Henikoff, D. Barbash, etc.

Thank you. We have already explained our responses to these concerns in comments above when addressing on the general criticisms of Reviewer 1. We have not duplicated the answer here.

Along the same line, if the rex sites have evolved so fast, one might expect that the proteins of the DCC have also evolved fast. The conservation between the orthologs between the two nematode species appears quite low (compared to other proteins) but this aspect of the comparison is not well documented.

As mentioned in response to comment 3 of reviewer 1, the revised manuscript includes a new figure that summarizes conservation of SDC-2 (Figure 1-—figure supplement 2A) and DPY-27 (Figure 1-—figure supplement 2B) in the *Caenorhabditis* genus. The additional description in that response is not duplicated here.

On Line #66 for instance, the authors mention that SDC-2 has "no homologs outside nematodes"..it seems it is also poorly conserved within nematodes. In general, it would be important to know the sequence conservation with other nematodes. Although I do not necessarily request a dN/dS analysis on the evolution of these proteins, having at least an idea of their conservation would be informative.

As explained above, please see the new Figure 1-—figure supplement 2A,B, which documents the conservation of SDC-2 across species.